# An Adaptive Sand Cat Swarm Algorithm Based on Cauchy Mutation and Optimal Neighborhood Disturbance Strategy

**DOI:** 10.3390/biomimetics8020191

**Published:** 2023-05-04

**Authors:** Xing Wang, Qian Liu, Li Zhang

**Affiliations:** 1School of Science, Xi’an University of Technology, Xi’an 710054, China; 2School of Science, Chang’an University, Xi’an 710064, China

**Keywords:** sand cat swarm optimization algorithm, nonlinear adaptive parameter, Cauchy mutation, optimal neighborhood disturbance, competition suite, engineering optimization problems

## Abstract

Sand cat swarm optimization algorithm (SCSO) keeps a potent and straightforward meta-heuristic algorithm derived from the distant sense of hearing of sand cats, which shows excellent performance in some large-scale optimization problems. However, the SCSO still has several disadvantages, including sluggish convergence, lower convergence precision, and the tendency to be trapped in the topical optimum. To escape these demerits, an adaptive sand cat swarm optimization algorithm based on Cauchy mutation and optimal neighborhood disturbance strategy (COSCSO) are provided in this study. First and foremost, the introduction of a nonlinear adaptive parameter in favor of scaling up the global search helps to retrieve the global optimum from a colossal search space, preventing it from being caught in a topical optimum. Secondly, the Cauchy mutation operator perturbs the search step, accelerating the convergence speed and improving the search efficiency. Finally, the optimal neighborhood disturbance strategy diversifies the population, broadens the search space, and enhances exploitation. To reveal the performance of COSCSO, it was compared with alternative algorithms in the CEC2017 and CEC2020 competition suites. Furthermore, COSCSO is further deployed to solve six engineering optimization problems. The experimental results reveal that the COSCSO is strongly competitive and capable of being deployed to solve some practical problems.

## 1. Introduction

Throughout history, optimization issues have been presented in all dimensions of people’s lives, such as in finance, science, engineering, etc. Nevertheless, with the development of society, optimization issues have become progressively more intricate. Traditional optimization methods, such as the Lagrange multiplier method, the complex method, queuing theory, and so on, require explicit descriptions of conditions and can only solve smaller optimization problems, which cannot be tackled exactly in a limited time. At the same time, for nonlinear engineering problems with a large quantity of constraints and decision variables, traditional optimization methods tend to get caught in the local optimum instead of sourcing the global optimal solution. Therefore, drawing inspiration from numerous manifestations in nature, researchers have devised a host of powerful and accessible meta-heuristic algorithms that, it is worth noting, can strike a superior balance between hopping out of the topical optimum and converging to a single point in order to arrive at a global optimum and solve sophisticated optimization problems.

The algorithms have been grouped I”to f’ve principal categories based on the inspiration used to create them: (1). Human-based optimization algorithms are designed based on human brain thinking, systems, organs, and social evolution. An example is the well-known neural network algorithm (NNA) [1], which tackles problems in ways informed by the message transmission of neural networks in the human brain. The Harmony Search (HS) [2,3] algorithm simulates a musician’s ability to achieve a pleasing harmonic state by repeatedly adjusting the pitch through recall. (2). Those that emulate natural evolution are classified as evolutionary-based optimization algorithms. The genetic algorithm (GA) [4] is the most classical model for simulating evolution, in which chromosomes pass through a cycle of stages to form descendants that leave more adaptive individuals through the laws and methods of superiority and inferiority. Simultaneously, the differential evolution (DE) [5,6] algorithm, imperial competition algorithm (ICA) [7], and memetic algorithm (MA) [8] also belong to the algorithms based on evolutionary mechanisms. (3). Population-based optimization algorithms are modeled to simulate the reproduction, predation, migration, and other behaviors of a colony of organisms. In this class of algorithm, the individuals in the population are conceived as quality-free particles seeking the best position. Ant colony optimization (ACO) [9,10] exploits the ideology of ants searching for the shortest distance from the nest to food. Particle swarm optimization (PSO) [11], stemming from the feeding of birds, is the most broadly accepted swarm intelligence algorithm. The moth-flame optimization (MFO) [12] algorithm serves as a mathematical model that is built by simulating the special navigation of a moth, which spirals close to a light source until it “flames”. Other swarm intelligence algorithms include the gray wolf optimization (GWO) [13] algorithm, manta ray foraging optimization (MRFO) algorithm [14], artificial hummingbird algorithm (AHA) [15], dwarf mongoose optimization (DMO) [16], and chimpanzee optimization algorithm (CHOA) [17,18], etc. (4). Plant growth-based optimization algorithms. Such algorithms are devised inspired by the properties of plants, such as photosynthesis, flower pollination, and seed dispersal. The dandelion optimization (DO) [19] algorithm is inspired by its process of rising, falling, and landing in different weather conditions depending on the wind. The one that simulates the aggressive invasion of weeds, searches for a suitable living space, and utilizes natural resources for rapidly growing and reproducing is denoted as invasive weed optimization (IWO) [20]. (5). Physics-based optimization algorithms are created in accordance with physical phenomena and regulations in nature. The gravitational search algorithm (GSA) [21], which is derived from gravity, has a robust global search capability and a fast convergence rate. The artificial raindrop optimization algorithm (ARA) [22] is designed on the basis of the processes of raindrop formation, landing, collision, confluence, and finally evaporation as water vapor.

Especially, many algorithms have been implemented for many practical engineering problems on account of their excellent performance, such as feature selection [23,24,25], image segmentation [26,27], signal processing [28], construction of water facilities [29], path planning for walking robots [30,31], job-shop scheduling problems [32], and piping and wiring problems in industrial and agricultural production [33]. Unlike gradient-based optimization algorithms, meta-heuristic algorithms rely on probabilistic search rather than gradient-based execution. With no centralized control constraints, the failure of individual individuals does not affect the solution of the whole problem, ensuring a more stable search process. As a general rule, as a first step, it is necessary to appropriately setup the essential parameters in the algorithm and produce a stochastic collection of initial solutions. Next, the search mechanism of the algorithm is applied to help find the optimum value until the stopping constraint is attained or the optimum value is discovered [34]. Nevertheless, it is evident that there are two different aspects to every algorithm; there are merits and demerits, and the performance will fluctuate based on the problem being addressed. No free lunch (NFL) [35] claims that an algorithm is capable of addressing one or more optimization problems, but there is no scientific foundation for the idea that it is possible to successfully tackle other optimization problems. Therefore, facing several special problems, it is sensible to propose a variety of strategies to enhance the efficiency of the algorithm.

The sand cat swarm optimization algorithm is a recently published, completely new swarm intelligence algorithm. In [36], SCSO is tested with some other popular algorithms (such as PSO and GWO) on different test functions, and better results or at least comparable results are achieved, but they can still be further improved. As a result, this paper provides an enhancement to tackle the optimization problem with the following primary contributions:(1)The COSCSO with better performance is designed by adding three strategies to SCSO.

In the first place, the nonlinear adaptive parameters replace the original linear parameters to increase the global search and prevent it from being caught in a topical optimum.

In second place, the Cauchy mutation operator strategy expedites the convergence speed.

In the end, optimal neighborhood disturbance enriches population diversity.

(2)The enhanced algorithm is instrumented on test suites of different dimensions and on real engineering optimization problems.

Analyzing the balance of COSCSO exploration and exploitation on the 30-dimensional CEC2019 test suite.

Comparing with other competitive algorithms on the CEC2017 test suite and the CEC2020 test suite of 30 and 50 dimensions.

The improved algorithm is deployed on six engineering optimization problems in conjunction with nine other algorithms.

The remaining details of the paper are described below. The second part describes the relevant work on SCSO, with the third part consisting of a summary review of the original algorithm for sand cat swarm search for attacking prey. The fourth part elaborates on the three improvement strategies in detail. The fifth part presents an analysis of the comparative data of COSCSO, SCSO, and other optimization algorithms, while the superiority of COSCSO is illustrated. In the sixth part, six engineering examples are collected to verify the capabilities of COSCSO with other algorithms in addressing real-world problems. The final part is the conclusion.

## 2. Related Works

Since the emergence of the sand cat swarm optimization algorithm, considerable attention has been paid to it by researchers due to its excellence. Vahid Tavakol Aghaei, Amir SeyyedAbbasi et al. [37] applied COSCSO to address three diverse nonlinear control systems for inverted pendulum, Furuta pendulum, and Acrobat robotic arm. It has been shown through simulation experiments that SCSO is simple and accessible and can be a viable candidate for real-world control and engineering problems. In addition, several researchers have optimized the SCSO for greater performance. Firstly, Li et al. [38] designed an elite collaboration strategy with stochastic variation to select the top three sand cats in the population for adaptation, and the three elites assigned different weights cooperated to form a new sand cat position to guide the search process, avoiding the dilemma of being entangled in a local optimum. Secondly, Amir Seyyedabbasi et al. [39] combined SCSO with reinforcement learning techniques to better balance the exploration and exploitation processes and further solve the mobile node localization problem in wireless sensor networks. Finally, the ISCSO proposed by Lu et al. [40] effectively boosts the fault diagnosis performance of power transformers.

## 3. The Sand Cat Swarm Optimization

The sand cat swarm optimization (SCSO) algorithm is a remarkably new meta-heuristic optimization algorithm proposed by Amir Seyyedabbasi et al. in 2022. Sand cats live in very barren deserts and mountainous areas. Gerbils, hares, snakes, and insects are their dominant sources of food. In appearance, sand cats are similar to domestic cats, but one big difference is that their hearing is very sensitive and they can detect low-frequency noise below 2 kHz. Therefore, they can use this special skill to find and attack their prey very quickly. The process from discovery to prey capture is shown in Figure 1. We can compare the sand cat’s predation to the process of finding the optimal value, which is the inspiration of the algorithm.

### 3.1. Initialization

Originally, it is initialized in a randomized manner so that the sand cats are evenly distributed in the exploration area:(1)$$X_{0}=lb+rand\left(0,1\right)\cdot (ub-lb)$$
where *lb* and *ub* are the upper and lower bounds of the variable, and *rand* is a random number between 0 and 1.

The resulting initial matrix is shown below:(2)$$Cat=\left[\begin{array}{cccc} x_{1,1} & x_{1,2} & \cdots & x_{1,M}\\ x_{2,1} & x_{2,2} & \cdots & x_{2,M}\\ \vdots & \vdots & \cdots & \vdots \\ x_{N,1} & x_{N,2} & \cdots & x_{N,M} \end{array}\right]$$
where $x_{i,j}$ denotes the *j*th dimension of the *i*th individual, and there are a total of *N* individuals and *M* variables. Meanwhile, the matrix of the fitness function is shown below:(3)$$Fitness=\left[\begin{array}{c} f(x_{1,1};x_{1,2};\cdots x_{1,M})\\ f(x_{2,1};x_{2,2};\cdots x_{2,M})\\ \vdots \\ f(x_{N,1};x_{N,2};\cdots x_{N,M}) \end{array}\right]$$

After comparing all fitness values, the minimum value is found, and the individual corresponding to it is the current optimal one.

### 3.2. Searching for Prey (Exploration)

The sand cat searches for prey mainly using its very sharp sense of hearing, which can detect low-frequency noise below 2 kHz. Then its mathematical model in the prey-finding stage is shown as follows:(4)$$S_{e}=S_{M}-(S_{M}\times \frac{t}{T})$$
(5)$$r_{e}=S_{e}\times rand\;(0,1)$$
(6)$$X(t+1)=r_{e}\cdot (X_{a}(t)-rand\;(0,1)\cdot X(t))$$
where $S_{M}=2$, $S_{e}$ denotes the general sensitivity range of the sand cats, whose value decreases linearly from 2 to 0, and *r_e_* is the sensitivity range of a particular sand cat in the sand cat swarm. *t* is the immediate count of the iteration, and *T* depicts the utmost count of iterations for the entire search process. $X_{a}(t)$ is any one of the populations, and X(t) is the immediate position of the sand cat. Notably, when $S_{e}=0$, $r_{e}=0$, the latest position of the sand cat will also be assigned to 0 according to Equation (6), also in the search space. Furthermore, in order to guarantee a steady state between the exploration and exploitation phases, *R_e_* is put forward, and $R_{e}\in \left[0,2\right]$, its value is given by Equation (7).
(7)$$R_{e}=2\times S_{e}\times rand\;(0,1)-S_{e}$$

### 3.3. Grabbing Prey (Exploitation)

As the search process progresses and the sand cat attacks the prey found in the previous stage, its mathematical modeling of the prey attack phase is as follows:(8)$$dist=\left|rand\;(0,1)\cdot X_{best}(t)-X(t)\right|$$
(9)$$X(t+1)=X(t)-dist\cdot \cos(\theta )\cdot r_{e}$$
where *dist* is the distance between the best and the current individual. θ is a random angle from 0 to 360.

### 3.4. Bridging Phase

The conversion of SCSO from the exploration phase to exploitation is closely associated with the parameter *R_e_*. When $\left|R_{e}\right|<1$, the sand cat gets in close and captures the prey, which is in the exploitation phase; when $\left|R_{e}\right|>1$, it continues to search different spaces to find the location of the prey, which is in the exploration phase. The pseudo-code of SCSO is seen in [36]. The mathematical modeling at this time is:(10)$$X(t+1)=\left\{\begin{array}{c} X_{best}(t)-dist\cdot \cos(\theta )\cdot r_{e},\\ r_{e}\cdot (X_{a}(t)-rand(0,1)\cdot X(t)), \end{array}\right.\begin{array}{c} \left|R_{e}\leq 1\right|\;;\mathrm{exploitation}\\ \left|R_{e}\geq 1\right|\;;\mathrm{exploration} \end{array}$$

## 4. Improved Sand Cat Swarm Optimization

In SCSO, the sand cat uses its powerful ability to recognize lower-profile noise below 2 kHz to capture prey, although the algorithm is straightforward and accessible to implement and allows for iterating quickly until the best position is found. However, there are some shortcomings, such as the tendency to be stuck in the topical optimum and excessive premature convergence. So now this algorithm is optimized and improved. In this paper, three strategies will be taken, namely: nonlinear adaptive parameter, Cauchy mutation strategy, and optimal neighborhood disturbance strategy.

### 4.1. Nonlinear Adaptive Parameters

In SCSO, the parameter *S_e_* plays a very prominent role; firstly, it indicates the sensitivity range of the sand cat hearing. Secondly, it influences the size of the parameter *R_e_*, which is in turn accountable for equilibrating the global search and local exploitation phases of the iterative process, and thus *S_e_* is also a parameter that coordinates the exploration and exploitation phases. Finally, it is also a crucial component of the convergence factor *r_e_*, which affects the speed of convergence during the iteration. Whereas in the original algorithm, *S_e_* decreases linearly from 2 to 0. This idealized law is not representative of the actual sand cat’s predation ability, so a nonlinear adaptive parameter strategy is now utilized with the formula as in Equation (11).
(11)$$S_{e}=2\left|q_{t}\right|\left\{1-{\left(\frac{t}{T}\right)}^{2}-\left[1-{\left(\frac{t}{T}\right)}^{{}^{\frac{1}{4}}}\right]\right\}+2\left[1-{\left(\frac{t}{T}\right)}^{\frac{1}{4}}\right]$$

Here, $q_{t}=1-2{\left(q_{t-1}\right)}^{2}$, and $q_{t}\in \left[0,1\right],q_{t}\neq 0.5$.

The variation curves before and after the improvement of parameter *S_e_* are displayed in Figure 2. Comparing the two curves, we can see that the modified *S_e_* has a larger value in the preliminary portion of the optimization process, focusing on the global search; moreover, due to the perturbation of *q_t_*, the value of *S_e_* sometimes becomes smaller in the optimization process, which can cater for the local search at this time, forming a faster convergence speed and enabling a more precise search accuracy. In the posterior part of the optimization process, the value is on the lower side, focusing on the local search, and due to the perturbation of *q_t_*, the value of *S_e_* sometimes becomes larger in the optimization process, which ensures that the algorithm avoids becoming bogged down in local optima.

### 4.2. Cauchy Mutation Strategy

The Cauchy distribution is distinguished by long tails at both ends and a larger peak at the central origin. The introduction of the Cauchy mutation operator [41,42,43] as a mutational step provides each sand cat with a greater likelihood of skipping to a better place. Once obtaining the local optimal solution, the Cauchy mutation operator perturbs the step size, making the step size larger, which in turn causes the sand cat to jump away from the local optimal position. Conversely, this operator makes the step size smaller and speeds up the convergence when the individual is pursuing the global optimum. The Cauchy mutation has been integrated with many algorithms, such as MFO and CSO. The Cauchy distribution function and the probability density function of the Cauchy distribution are described as follows:(12)$$F(x)=\frac{1}{\pi }\arctan(\frac{x-x_{0}}{\gamma })+\frac{1}{2}$$
(13)$$f(x)=\frac{1}{\pi }\left[\frac{\gamma }{{\left(x-x_{0}\right)}^{2}+\gamma^{2}}\right]$$
where *x_0_* is referred to as the position parameter at the maximum and *γ* is the size parameter of half the distance at half the width of the peak. Here $x_{0}=0,\gamma =1$, the standard Cauchy distribution is obtained, and its probability density function is as in Equation (14), and Figure 3 is the probability density function curve of the standard Cauchy distribution.
(14)$$f(x)=\frac{1}{\pi (1+x^{2})}$$

To diminish the probability of dropping into the local optimum of SCSO, this paper uses the Cauchy mutation operator to promote the global optimization-seeking ability of the algorithm, expedite the convergence speed, and increase the population diversity. Well, at this point, the individual renewal changes to
(15)$$X(t+1)=X(t)-C(0,1)\cdot r_{e}\cdot dist\cdot \cos(\theta )$$
where C(0,1) is a stochastic number that submits to the standard Cauchy distribution.

### 4.3. Optimal Neighborhood Disturbance Strategy

When a sand cat swarm is feeding, all individuals move towards the location of prey, a circumstance that may account for the homogeneity of the population but is not conducive to the fluidity of the global search phase. Therefore, an optimal neighborhood disturbance strategy [44] is now utilized. When the global optimum is updated, a further search is performed around it. With this, population diversity can be enriched to obviate the need for a local optimum. The optimal neighborhood disturbance is shown as follows:(16)$$X_{best}^{*}(t)=\left\{\begin{array}{c} X_{best}(t)+0.5\cdot r_{1}\cdot X_{best}(t),\\ X_{best}(t), \end{array}\right.\begin{array}{c} r_{2}<0.5\\ r_{2}\geq 0.5 \end{array}$$
where $X_{best}^{*}(t)$ is the new individual generated after disturbance, $r_{1},r_{2}\in [0,1]$.

After the optimal neighborhood search, the greedy strategy is adopted to opt for judgment. The specific formula is as follows:(17)$$X_{best}(t)=\left\{\begin{array}{c} X_{best}^{*}(t),\\ X_{best}(t), \end{array}\right.\begin{array}{c} f(X_{best}^{*}(t))<f(X_{best}(t))\\ f(X_{best}(t))\leq f(X_{best}^{*}(t)) \end{array}$$

### 4.4. COSCSO Steps

In this work, a nonlinear adaptive parameter, a Cauchy mutation strategy, and an optimal neighborhood disturbance strategy are combined to modify the standard SCSO algorithm to form the COSCSO algorithm. The fundamental steps of COSCSO are as follows:

**Step 1.** Initialization, identifying the population magnitude *N*, the maximum number of iterations *T*, and the parameters needed.

**Step 2.** Computing and comparing the fitness value of each sand cat and getting the existing best position.

**Step 3.** Update the nonlinear parameters *S_e_* and the parameters *r_e_*, *R_e_* by means of Equations (11), (5) and (7).

**Step 4.** Generate the Cauchy mutation operator.

**Step 5.** Update the individual position of the sand cat if $\left|R_{e}\right|>1$, using Equation (6); otherwise, use Equation (15).

**Step 6.** Compare the fitness values of the existing individual, and if the former is better, renew the best individual position.

**Step 7.** Generate new individuals by perturbing the existing best individual according to the optimal neighborhood disturbance strategy using Equation (16).

**Step 8.** A comparison of the fitness values of the freshly engendered individual and the best individual in accordance with the greedy strategy, and upgrading the position of the best individual if the former is preferable.

**Step 9.** Revert to Step 3 if the maximum count of iteration *T* has not been achieved; otherwise, continue with Step 10.

**Step 10.** Output the global best position and the corresponding fitness value.

For a more concise description of the procedures of the COSCSO algorithm, the pseudo-code of the algorithm is given in Table 1 and the flowchart in Figure 4.

### 4.5. Computational Complexity of COSCSO Algorithm

The computational complexity of an algorithm is defined as the volume of resources it consumes during implementation. When the COSCSO algorithm program is performed, the complexity of each *D*-dimensional individual in the population is O(*D*). Then, for a population size of *N* individuals, its computational complexity is O(*N × D*), and in the process of finding the best, it needs to be executed *T* times to get the final result, and the final is O(*T × N × D*). In the following section, we will test the capability of COSCSO by exploiting different test suites and concrete engineering problems.

## 5. Numerical Experiments and Analysis

In this chapter, the balance between the COSCSO exploration and development processes is first discussed. Then, the more challenging CEC2017 test suite and the CEC2020 test suite were selected to test the final performance of COSCSO. COSCSO is evaluated with standard SCSO as well as with an extensive variety of meta-heuristic algorithms, and the values of the required parameters for all algorithms are specified in Table 2. All statistical experiments are conducted on the same computer. In addition, all algorithms are implemented in 20 independent executions of each function, taking *N* = 50 and *T* = 1000. And the optimization results are compared by analyzing the average and standard deviation of the best solutions.

### 5.1. Exploration and Exploitation Analysis

Exploration and exploitation play an integral role in the optimization process. Therefore, when evaluating algorithm performance, it is vital to discuss not only the ultimate consequences of the algorithm but also the nature of the balance between exploration and exploitation [45]. Figure 5 gives a diagram of the exploration and exploitation of COSCSO on the 30-dimensional CEC2020 test suite.

As we can observe from the figure, the algorithm progressively transitions from the exploration phase to the exploitation phase. On the simpler basic functions F2 and F4 and the most complex composition function F9, COSCSO moves to the exploitation phase around the 10th iteration and rapidly reaches the top of the exploitation phase, illustrating the greatly enhanced convergence accuracy of COSCSO. On the hybrid functions F5, F6, and F7, COSCSO also preserves a strong exploration ability in the middle and late stages, effectively refraining from plunging into a local optimum.

### 5.2. Comparison and Analysis on the CEC2017 Test Suite

Firstly, a running test is performed on the 30-dimensional CEC2017 test suite. The specific formulas for these functions are given in [46]. Then, COSCSO is compared and analyzed with SCSO and eight other competitive optimization algorithms, which include: PSO, RSA [47], BWO [48], DO, AOA [49], HHO [50,51], NCHHO [52], and ATOA [53].

The results obtained by running COSCSO 20 times with other competing algorithms are given in Table 3. There are 24 test functions ranked first in COSCSO, accounting for about 82.76% of all test functions. At first, on single-peak test functions, COSCSO has a distinct superiority over others in regard to the mean value and can achieve a smaller standard deviation. Next, on the multi-peak test function, although COSCSO is at a weak point compared to PSO in F5 and F6, it is more competitive with the other nine algorithms. Furthermore, on the hybrid functions, except for F15 and F19, COSCSO is obviously superior to other algorithms, especially on F12–F14, F16, and F18, where COSCSO is on the leading edge with respect to mean and standard deviation. Finally, on the synthetic functions, COSCSO is far ahead on F22, F28, and F30, but on F21, it is marginally weaker than PSO and SCSO. The last row of the table shows the average ranking of the ten algorithms. The rankings are: COSCSO > HHO > SCSO > PSO > DO > ATOA > AOA > NCHHO = BWO > RSA. In summary, the COSCSO algorithm has superior merit-seeking ability on the CEC2017 test suite; this fully demonstrates that the three strategies effectively boost convergence accuracy and efficiency and greatly reduce the defects of the initial algorithm.

Table 3 depicts the Wilcoxon rank-sum test *p*-values [54] derived from solving the 30-dimensional CEC2017 problem for 20 runs of other meta-heuristic algorithms at the 95% significance level (α=0.05), using COSCSO as a benchmark. The last row shows the statistical results, “+” indicates the number of algorithms that outperform the COSCSO, and “=” indicates that there is no appreciable variation among the two algorithms, at this point α=0.05. “-” indicates the number of times COSCSO outperformed other algorithms. Combining the ranking of each algorithm, we get that COSCSO is significantly superior to RSA, BWO, DO, AOA, NCHHO, and ATOA on all test functions, worse than PSO on F6 and F21, and apparently preferred to PSO on 14 test functions. So, all together, COSCSO has by far better competence compared to other algorithms and is a wise choice for solving the CEC2017 problem.

Figure 6 illustrates the convergence curves of COSCSO with other algorithms on the CEC2017 test functions. Observing the curves, we can see that COSCSO is a dramatic enhancement over SCSO. Although for F5, F6, and F21, COSCSO is at a disadvantage compared to PSO and inferior to the ATOA on F15 and F19, COSCSO is still more superior than the other algorithms. On the remaining functions, COSCSO obviously converges faster and with higher convergence accuracy than SCSO. These advantages are attributed to the improvement of three major strategies of adaptive parameters, Cauchy mutation operator and optimal neighborhood disturbance, which hinder the algorithm from dropping into local optimum and excessive premature convergence.

Figure 7 depicts the box plots of COSCSO with other algorithms on the CEC2017 test functions. The height of the box mirrors the level of swing in the data, and a narrower box plot represents more concentrated data and a more stable algorithm. If there are abnormal points in the data that are beyond the normal range of the data, these points are signaled by a “+”. From the figure, we can see that on F1, F3, F4, F11, F12, F14, F15, F17, F18, F27, F28, and F30, the box plot width of the COSCSO is significantly narrower than other algorithms. In addition, except for F22, the COSCSO has almost no outliers. This implies that its operation is more stable and has good robustness in solving the CEC2017 test functions.

Radar maps, also known as spider web maps, map the amount of data in multiple dimensions onto the axes and can give an indication of how high or low the weights of each variable are. Figure 8 shows the radar maps of COSCSO with other algorithms, which are plotted based on the ranking of the ten meta-heuristic algorithms on the CEC2017 test function. From the figure, it can be observed that COSCSO constitutes the smallest shaded area, which further sufficiently illustrates the capacity of COSCSO ahead of the other nine comparative algorithms. The shaded area of HHO ranks second, which indicates that HHO has some competition for COSCSO.

### 5.3. Comparison and Analysis on the CEC2020 Test Suite

In order to further test the COSCSO’s optimization-seeking ability, this paper is also tested on the 30-dimensional and 50-dimensional CEC2020 test suites, respectively. The CEC2020 test suite [55] is composed of some of the CEC2014 test suite [56] and the CEC2017 test suite. The algorithms compared with it are eight other optimization algorithms besides SCSO, which include WOA [57], RSA, PSO, CHOA, AOA, HHO, NCHHO, and ATOA. All parameter definitions remain identical except for the number of dimensions.

The experimental results of each algorithm on the 30-dimensional CEC2020 test suite are given in Table 4. From the data, it can be seen that COSCSO is ahead of SCSO and other comparative algorithms on nine test functions. And on F6, the HHO ranks first and the COSCSO ranks second, which is better than the other eight algorithms. The smallest standard deviation on F1, F5, and F7 indicates that COSCSO is more steady on these test functions. The table shows that the overall ranking is COSCSO > HHO > SCSO > PSO > WOA > ATOA > CHOA > AOA > RSA > NCHHO. The average rank of COSCSO is 1.1, which is the first overall rank, and the average rank of HHO is 2.8, which is the second overall rank, which shows that COSCSO is consistently first among all algorithms many times.

In addition, Table 4 lists the *p*-value magnitude of each algorithm, from which it can be seen that COSCSO as a whole outperforms all compared algorithms, especially for the WOA, RSA, PSO, CHOA, AOA, NCHHO, and ATOA, the COSCSO algorithms far ahead. For the HHO and SCSO, there is no major difference in a few test functions. This reveals that COSCSO is extremely feasible for solving the CEC2020 function problem in 30 dimensions.

Figure 9 presents the convergence curves of COSCSO with other algorithms on the 30-dimensional CEC2020 test suite. Combining the data in the table visually illustrates that COSCSO has faster convergence and more accurate accuracy on F1, F2, F5, F7, and F8. It is poorer than the HHO on F6.

Figure 10 displays the box plots of COSCSO with other algorithms on the 30-dimensional CEC2020 test function. Where the COSCSO algorithm has the smallest median on F1, F2, F5, F7, and F8 compared to the other nine algorithms. In the plots of F1, F5, F7, F8, and F10, the box plot of COSCSO is narrower, suggesting that the COSCSO algorithm is more stable and has relatively good robustness on these functions.

Figure 11 presents the radar maps based on the ranking of the COSCSO with the other nine algorithms in the 30-dimensional CEC2020 test suite. Depending on the area of the radar maps, it is easy to see that COSCSO ranks at the top in all functions, which very intuitively shows the superiority of COSCSO and its applicability in solving the 30-dimensional CEC2020 problem.

Table 5 contains the experimental data for each algorithm for each metric on the 50-dimensional CEC2020 test function. In this experiment, COSCSO achieved better fitness values on the eight test functions. Although inferior to the original algorithm in F2 and F3, the COSCSO algorithm performed competitively compared to the other eight algorithms. The third row from the bottom is the average rank of the ten algorithms. COSCSO has an average rank of 1.4, ranking first. The combined ranking of the algorithms is: COSCSO > SCSO > HHO > PSO > WOA > ATOA > CHOA > RSA > AOA > NCHHO. This fully reflects the ability of the COSCSO algorithm to solve the CEC2020 problem.

Rank sum tests are also documented in Table 5. Similarly, the COSCSO was used as a benchmark, and other meta-heuristic algorithms were run 20 times to solve the 50-dimensional CEC2020 problem at the 95% significance level (α=0.05). Looking at the last row, COSCSO clearly excelled SCSO on the six tested functions; moreover, COSCSO outperformed the other algorithms on most tested functions.

The convergence plots of each function in Figure 12 more directly show its performance in solving the CEC2020 problem. COSCSO surpasses all other algorithms except F2, F3, and F4 and ranks first.

In Figure 13, the median is the same for all algorithms except the PSO algorithm on F4. The median of COSCSO is lower than the other algorithms except for F3, F6, and F9. The box plots of COSCSO on F1, F5, F7, and F10 are extremely narrow, indicating its good stability and robustness.

Figure 14 shows the radar maps of COSCSO with other algorithms. Observing the area of each graph, it can be detected that the shaded area of COSCSO is the smallest as well as relatively more rounded, which indicates that COSCSO has more stable and remarkable capability, and COSCSO can be deployed to solve the 50-dimensional CEC2020 problem.

## 6. Engineering Problems

This chapter tests the ability of COSCSO to solve practical problems [58]. In the following, ten algorithms are devoted to addressing six practical engineering problems: welded beam, pressure vessel, gas transmission compressor, heat exchanger, tubular column, and piston lever design problems. In particular, the bounded problems are converted into unbounded problems by utilizing penalty functions. In the comparison experiments, N=30, T=500, and running times are set to 20.

### 6.1. Welded Beam Design

The objective of the problem is to construct a welded beam [59] with minimal expense under the bounds of shear stress (η), bending stress (λ), buckling load ($Q_{C}$) and end deflection (μ) of the beam. It considers the weld thickness *h*, the joint length *l*, the height *t* of the beam, and the thickness *b* as variants, and the design schematic is shown in Figure 15. Let $K=[k_{1},k_{2},k_{3},k_{4}]=[h,l,t,b]$, the mathematical model of this problem is shown in Equation (18).
(18)$$\min\;f(K)=1.10471k_{1}^{2}k_{2}+0.04811k_{3}k_{4}(14.0+k_{2}),$$

Subject to:$$\begin{array}{l} y_{1}(K)=\eta (k)-\eta_{\max}\leq 0,\;y_{2}(K)=\beta (k)-\beta \leq_{\max}0,\;y_{3}(K)=\mu (k)-\mu_{\max}\leq 0,\\ y_{4}(K)=k_{1}-k_{4}\leq 0,\;y_{5}(K)=M-Q_{C}(k)\leq 0,\;y_{6}(K)=0.125-k_{1}\leq 0,\\ y_{7}(K)=1.1047k_{1}^{2}+0.04811k_{3}k_{4}(14.0+k_{2})-5.0\leq 0, \end{array}$$

Variable range:$$0.1\leq k_{1}\leq 2,0.1\leq k_{2}\leq 10,0.1\leq k_{3}\leq 10,0.1\leq k_{4}\leq 2,$$
where
$$\begin{array}{l} \eta (K)=\sqrt{{\left(\eta^{\prime }\right)}^{2}+2\eta^{\prime }\eta^{\prime \prime }\frac{k_{2}}{R}+{\left(\eta^{\prime \prime }\right)}^{2}},\;\eta^{\prime }=\frac{M}{\sqrt{2}k_{1}k_{2}},\;\eta^{\prime \prime }=\frac{WR}{J},\;W=M(S+\frac{k_{2}}{2}),\\ R=\sqrt{\frac{k_{2}^{2}}{4}+{\left(\frac{k_{1}+k_{2}}{2}\right)}^{2}},\;J=2\{\sqrt{2}k_{1}k_{2}[\frac{k_{2}^{2}}{4}+{\left(\frac{k_{1}+k_{3}}{2}\right)}^{2}]\},\;\beta (K)=\frac{6MS}{k_{4}k_{3}^{2}},\\ \mu (K)=\frac{6ML^{3}}{Ek_{4}k_{3}^{2}},\;Q_{C}(K)=\frac{4.013D\sqrt{\frac{k_{3}^{2}k_{4}^{6}}{36}}}{S^{2}}(1-\frac{k_{3}}{2S}\sqrt{\frac{D}{4G}}),\\ M=6000\;lb,\;S=14\;in,\;\mu_{\max}=0.25\;in,\;D=30\times 1^{6}\;psi,\\ G=12\times 10^{6}\;psi,\;\xi_{\max}=13,600\;psi,\;\beta_{\max}=30,000\;psi. \end{array}$$

Ten competitive meta-heuristic algorithms are used to solve this problem in this experiment, which are: COSCSO, SCSO, WOA, AO [60], SCA [61], RSA, HS, BWO, HHO, and AOA. The optimal cost obtained by solving the welded beam design problem using each algorithm and the decision variables it corresponds to are given in Table 6. It is apparent from the table that COSCSO generates the cheapest expenses. Table 7 shows the statistical results obtained for all algorithms run 20 times. It can be noticed that COSCSO obtained the best ranking in all indicators. In conclusion, COSCSO is highly competitive in solving the welded beam design problem.

### 6.2. Pressure Vessel Design

The main purpose of the problem is to fabricate the pressure vessel [62] with the least amount of cost under a host of constraints. It treats shell thickness *T*_1_, head thickness *T*_2_, inner radius *R**, and the length *S* of the cylindrical part without head as variables, and let $K=[k_{1},k_{2},k_{3},k_{4}]=[T_{1},T_{2},R^{*},S]$. The design schematic is presented in Figure 16. The mathematical model of the problem is shown in Equation (19).
(19)$$\min\;f(K)=0.6224k_{1}k_{3}k_{4}+1.7781k_{2}k_{3}^{2}+3.1661k_{1}^{4}+19.84k_{1}^{2}k_{3},$$

Subject to:$$\begin{array}{l} y_{1}(K)=-k_{1}-0.0193k_{3}\leq 0,\;y_{2}(K)=-k_{3}+0.00954k_{3}\leq 0,\\ y_{3}(K)=-\pi k_{3}^{2}k_{4}-4/3\pi k_{3}^{3}+1296000\leq 0,\;y_{4}(K)=k_{4}-240\leq 0, \end{array}$$

Variable range:$$1\times 0.0625\leq k_{1},k_{2}\leq 99\times 0.0625,\;10\leq k_{3},k_{4}\leq 200.$$

This problem is solved by ten algorithms, which are COSCSO, SCSO, WOA, AO, HS, RSA, SCA, BWO, BSA [63], and AOA. Table 8 contains the optimal cost of COSCSO and other compared algorithms and their corresponding decision variables. Four more pieces of data for each algorithm are included in Table 9. The result of COSCSO is the best among the ten algorithms and is relatively stable.

### 6.3. Gas Transmission Compressor Design Problem

The key target of the problem [64] is to minimize the total expense of carrying 100 million cubic feet per day. There are three design variables in this problem: the distance between the two compressors (*L*), the ratio of the first compressor to the second compressor pressure (δ), and the length of the natural gas pipeline inside the diameter (*H*). The gas transmission compressor is shown in Figure 17. Let $K=[k_{1},k_{2},k_{3}]=[L,\delta ,H]$. It is modeled as illustrated in Equation (20).
(20)$$\begin{array}{l} \min\;f(K)=3.69\times 10^{4}k_{3}+7.72\times 10^{8}k_{1}^{-1}k_{2}^{0.219}\\ \;\;\;\;-765.43\times 10^{6}k_{1}^{-1}+8.61\times 10^{5}\times k_{1}^{-\frac{1}{2}}k_{2}{\left(k_{2}^{2}-1\right)}^{-\frac{1}{2}}k_{3}^{-\frac{2}{3}}, \end{array}$$

Variable range:$$10\leq k_{1}\leq 55,\;1.1\leq k_{2}\leq 2,\;10\leq k_{3}\leq 40.$$

In addition to SCSO, we pick RSA, BWO, SOA [65], WOA, SCA, HS, AO, and AOA to compare with COSCSO. The best results of different algorithms and the corresponding decision variables are summarized in Table 10. The best results of COSCSO are substantially smaller than those of the other algorithms. The statistical results of all algorithms are collected in Table 11, where their standard deviations are the smallest, indicating a high stability of COSCSO.

### 6.4. Heat Exchanger Design

It is a minimization problem for heat exchanger design [66]. There are eight variables and six inequality constraints in this problem. It is specified as shown in Equation (21).
(21)$$\min\;f(K)=k_{1}+k_{2}+k_{3},$$

Subject to:$$\begin{array}{l} y_{1}(K)=0.0025(k_{4}+k_{6})-1\leq 0,\\ y_{2}(K)=0.0025(k_{5}+k_{7}-k_{4})-1\leq 0,\\ y_{3}(K)=1-0.01(k_{8}-k_{5})\geq 0,\\ y_{4}(K)=k_{1}k_{6}-833.33252k_{4}-100k_{1}+83,333.333\geq 0,\\ y_{5}(K)=k_{2}k_{7}-1250k_{5}-k_{2}k_{4}+1250k_{4}\geq 0,\\ y_{6}(K)=k_{3}k_{8}-k_{3}k_{5}+2500k_{5}-1,250,000\geq 0, \end{array}$$

Variable range:$$100\leq k_{1}\leq 10,000,1000\leq k_{2},k_{3}\leq 10,000,\;10\leq k_{i}\leq 1000\;(i=4,\ldots ,8).$$

For this problem, nine algorithms, such as WOA and HHO, are compared with COSCSO. Table 12 counts the best results of COSCSO and other algorithms and the best decision variables corresponding to them. The results of each algorithm are listed in Table 13. Apparently, the COSCSO algorithm obtains better results and is very competitive among all ten algorithms.

### 6.5. Tubular Column Design

The goal of this problem is to minimize the expense of designing a tubular column [67] to bear compressive loads under six constraints. It contains two decision variables: the average diameter of the column (*D*), and the thickness of the tube (*b*), let $K=[k_{1},k_{2}]=[D,b]$. Its design schematic is depicted in Figure 18. The model of this problem is indicated in Equation (22).
(22)$$\min\;f(K)=9.8k_{1}k_{2}+2k_{1},$$

Subject to:$$\begin{array}{l} y_{1}(K)=\frac{Q}{\pi k_{1}k_{2}\delta_{y}}-1,\;y_{2}(K)=\frac{8QH^{2}}{\pi^{3}Ek_{1}k_{2}(k_{1}^{2}+k_{2}^{2})}-1,\\ y_{3}(K)=\frac{2.0}{k_{1}}-1,\;y_{4}(K)=\frac{k_{1}}{14}-1,\\ y_{5}(K)=\frac{0.2}{k_{2}}-1,\;y_{6}(K)=\frac{k_{2}}{8}-1, \end{array}$$

Variable range:$$2\leq k_{1}\leq 14,\;0.2\leq k_{2}\leq 0.8,$$
where
$$\delta_{y}=500,\;E=0.84\times 10^{6}.$$

Table 14 shows the optimal costs and variables for COSCSO and the other nine algorithms. Observing the four indicators in Table 15, COSCSO obtained better values for all of them.

### 6.6. Piston Lever Design

The primary goal of the problem [68] is to minimize the amount of oil consumed when the piston lever is tilted from 0° to 45° under four constraints, thus determining *H*, *B*, *D*, and *K*. The schematic is seen in Figure 19. The mathematical expression of the problem is Equation (23).
(23)$$\min\;f(K)=\frac{1}{4}\pi k_{3}^{2}(L_{2}-L_{1}),$$

Subject to:$$\begin{array}{l} y_{1}(K)=MA\cos\theta -RF\leq 0\;\theta =45{}^{\circ},\\ y_{2}(K)=M(A-k_{4})-N_{\max}\leq 0,\\ y_{3}(K)=1.2(A_{2}-A_{1})-A_{1}\leq 0,\\ y_{4}(K)=k_{3}/3-k_{2}\leq 0, \end{array}$$
where
$$\begin{array}{l} R=\frac{\left|-k_{4}(k_{4}\sin\theta +k_{1})+k_{1}(k_{2}-k_{4}\cos\theta )\right|}{\sqrt{{\left(k_{4}-k_{2}\right)}^{2}+k_{1}^{2}}},\\ F=\pi Ck_{3}^{2}/4,\\ A_{1}=\sqrt{{\left(k_{4}-k_{2}\right)}^{2}+k_{1}^{2}},\\ A_{2}=\sqrt{{\left(k_{4}\sin45+k_{1}\right)}^{2}+{\left(k_{2}-k_{4}\cos45\right)}^{2}}\\ M=10,000\;\mathrm{lbs},C=1500\;\mathrm{psi},\\ A=240\;\mathrm{in},\;N_{\max}=1.8\times 10\;\mathrm{lbs}\;\mathrm{in}. \end{array}$$

Besides COSCSO and SCSO, SOA, MVO [69], HHO, etc., were also enrolled in the experiment. By looking at Table 16 and Table 17, COSCSO is the best choice within each of these ten algorithms to solve this problem.

## 7. Conclusions and Future Work

In this paper, SCSO based on adaptive parameters, Cauchy mutation, and an optimal neighborhood disturbance strategy are proposed. The nonlinear adaptive parameter replaces the linear adaptive parameter and increases the global search, which helps prevent premature convergence and puts exploration and development in a more balanced state. The introduction of the Cauchy mutation operator perturbs the search step to speed up convergence and improve search efficiency. The optimal neighborhood disturbance strategy is used to enrich the species diversity and prevent the algorithm from getting into the dilemma of the local optimum. COSCSO is evaluated against the standard SCSO and other challenging swarm intelligence optimization algorithms at CEC2017 and CEC2020 in distinct dimensions. The comparison of average and standard deviation, convergence, stability, and statistical analysis were performed. It is proven that COSCSO converges more rapidly, with higher accuracy, and stays more stable. In contrast to other algorithms, COSCO is more advanced. What is more, COSCSO is deployed to solve six engineering problems. From the experimental results, it can be visually concluded that COSCSO also has the potential to solve practical problems.

The COSCSO algorithm has strong exploration ability, which can effectively avoid falling into local optimums and prevent premature convergence. However, it has weak exploitation ability and a relatively slow convergence speed. In the future, we can use more novel strategies to improve the algorithm and further improve its exploration speed, which can be made available to tackle more high-dimensional optimization problems and employed in various fields, such as feature selection, path planning, image segmentation, fuzzy recognition, etc.

## Figures and Tables

**Figure 1 biomimetics-08-00191-f001:**
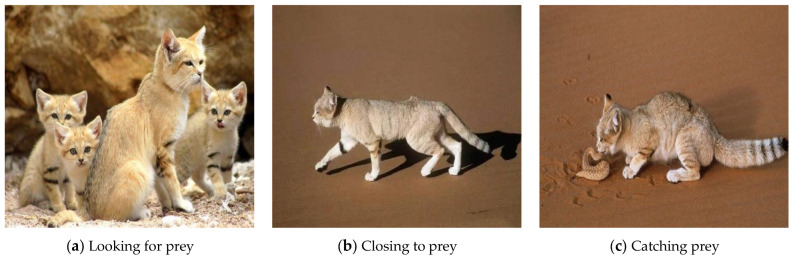
Sand cat capturing prey diagram.

**Figure 2 biomimetics-08-00191-f002:**
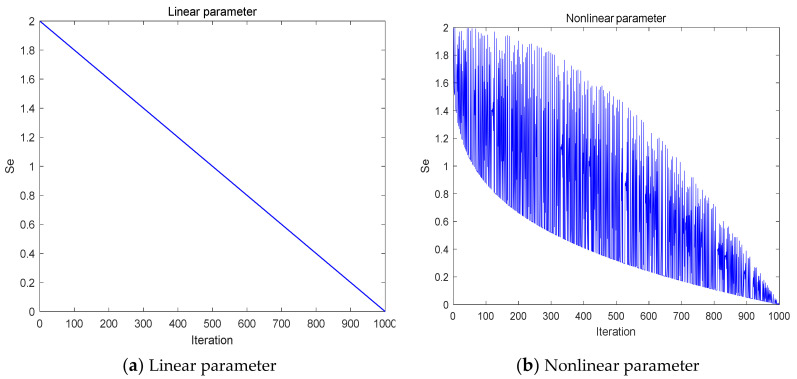
The curve of the variation of parameter *S_e_*.

**Figure 3 biomimetics-08-00191-f003:**
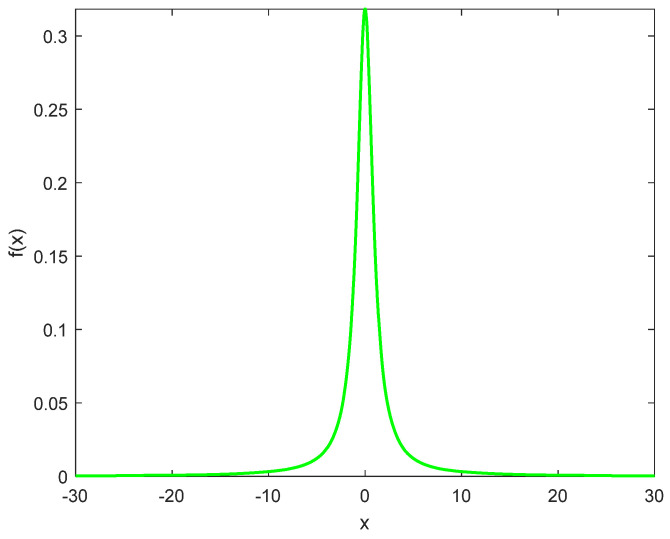
Curve of the probability density function of the Cauchy distribution.

**Figure 4 biomimetics-08-00191-f004:**
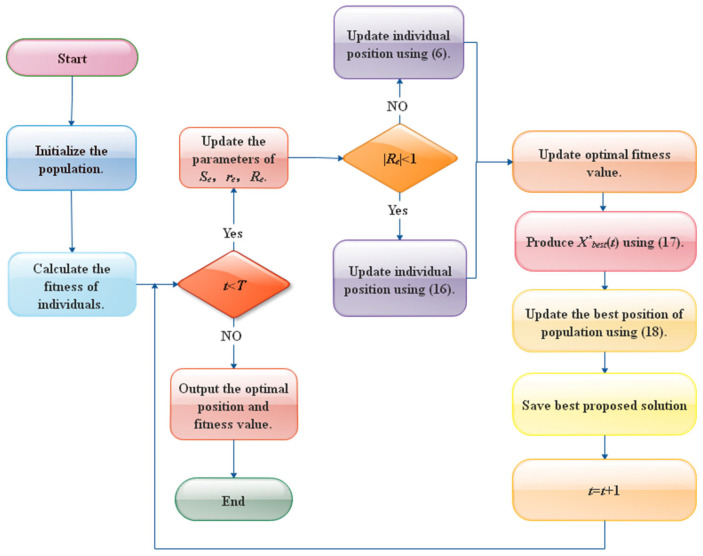
Flow chart of the COSCSO algorithm.

**Figure 5 biomimetics-08-00191-f005:**
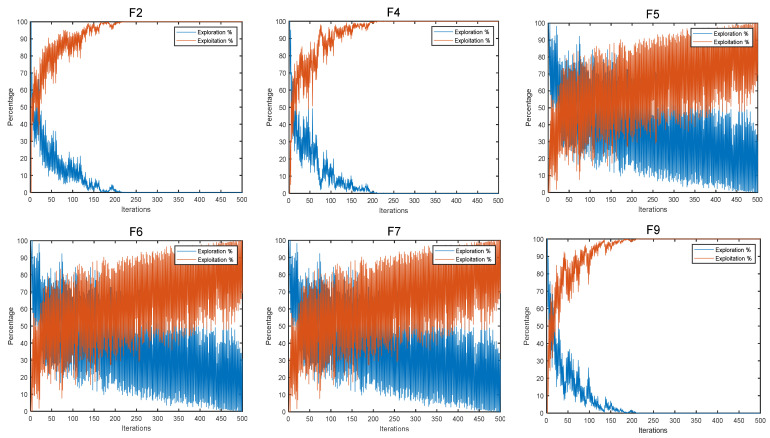
Diagram of COSCSO exploration and exploitation.

**Figure 6 biomimetics-08-00191-f006:**
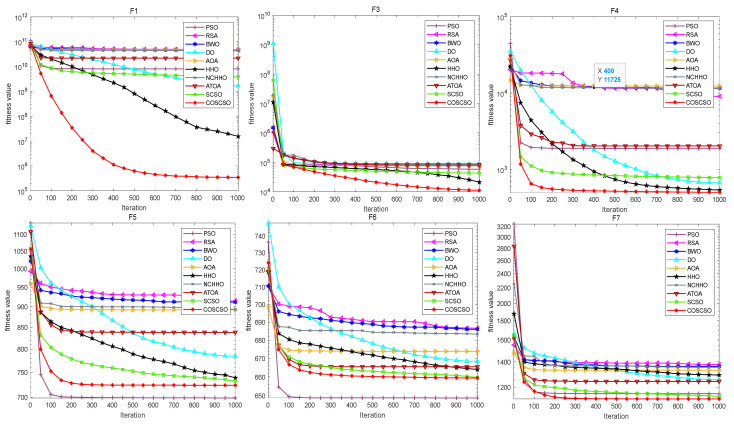
Convergence curves of COSCSO with other algorithms (CEC2017).

**Figure 7 biomimetics-08-00191-f007:**
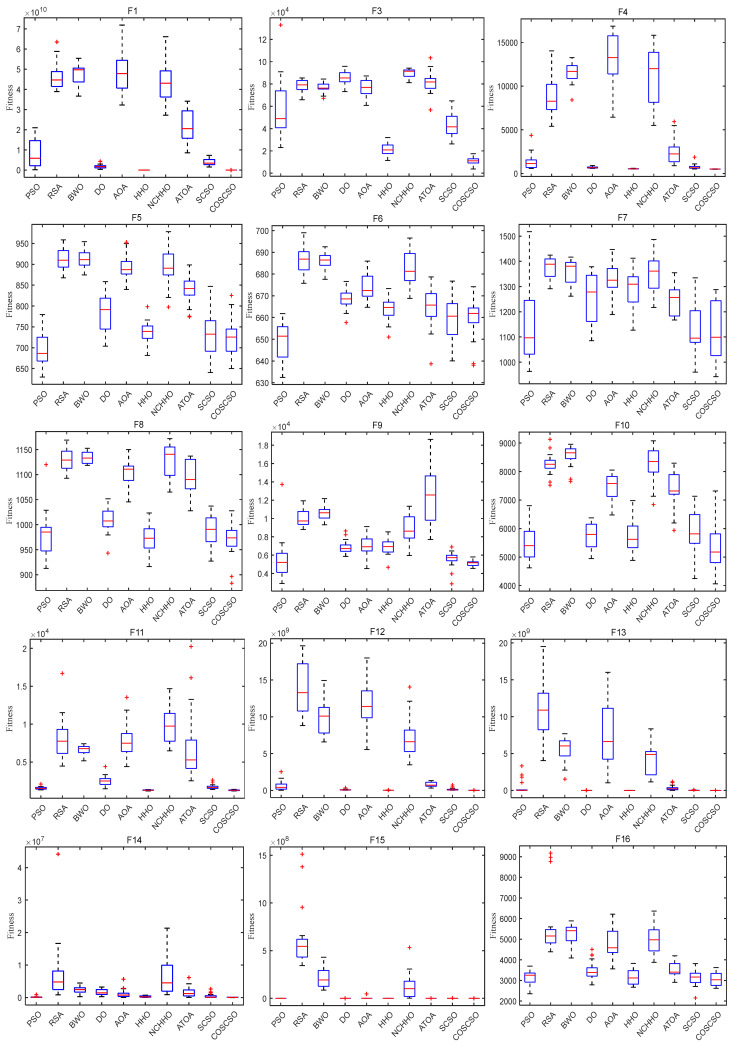
Box plots of COSCSO with other algorithms (CEC2017).

**Figure 8 biomimetics-08-00191-f008:**
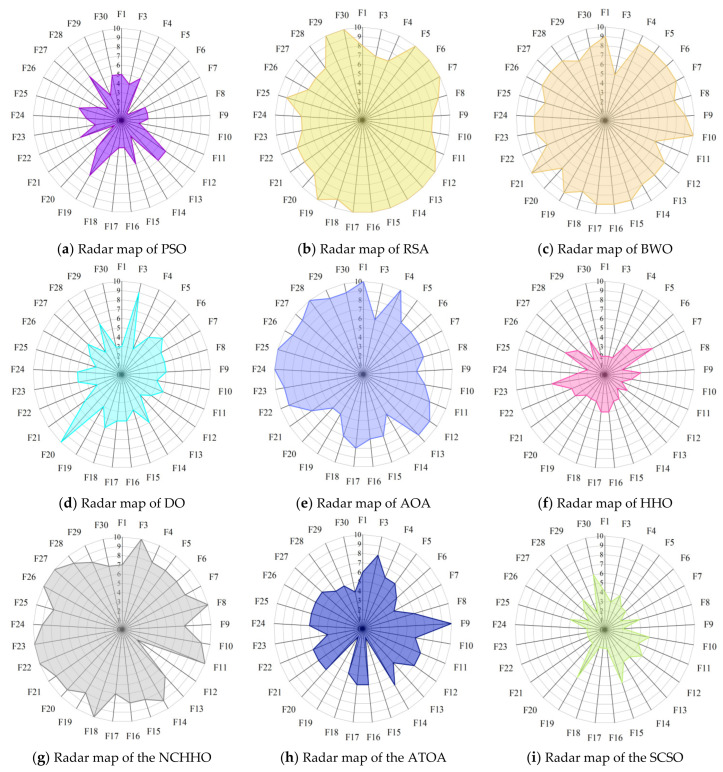
Radar maps of COSCSO with other algorithms (CEC2017).

**Figure 9 biomimetics-08-00191-f009:**
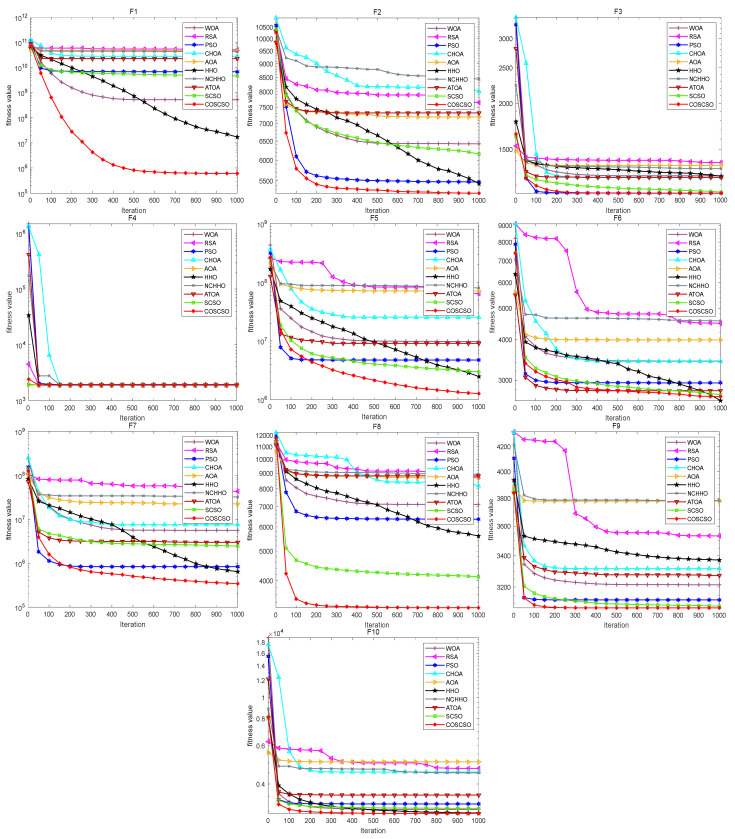
Convergence curves of COSCSO with other algorithms (30-dimensional CEC2020).

**Figure 10 biomimetics-08-00191-f010:**
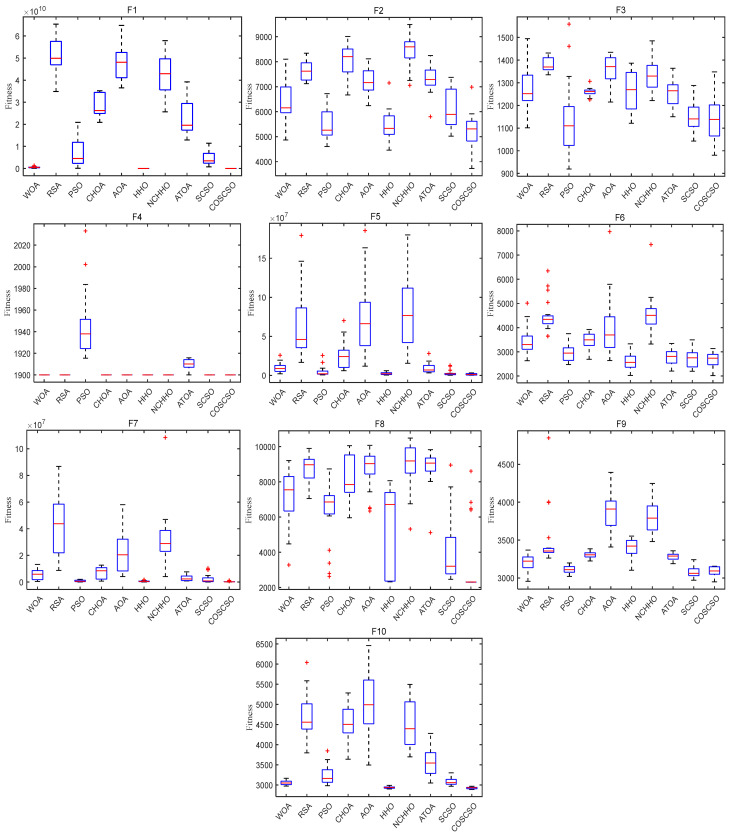
Box plots of COSCSO with other algorithms (30-dimensional CEC2020).

**Figure 11 biomimetics-08-00191-f011:**
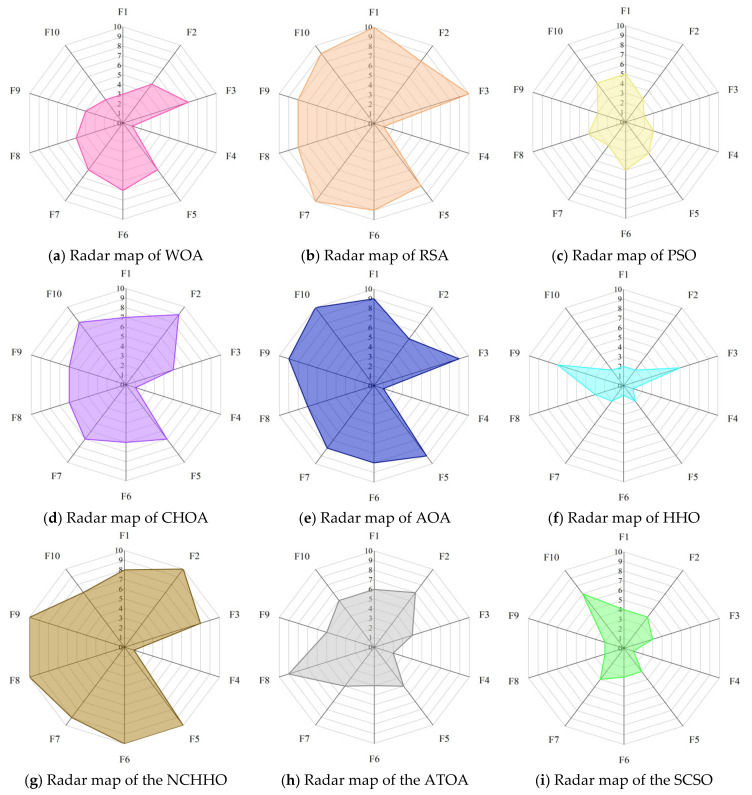
Radar maps of COSCSO with other algorithms (30-dimensional CEC2020).

**Figure 12 biomimetics-08-00191-f012:**
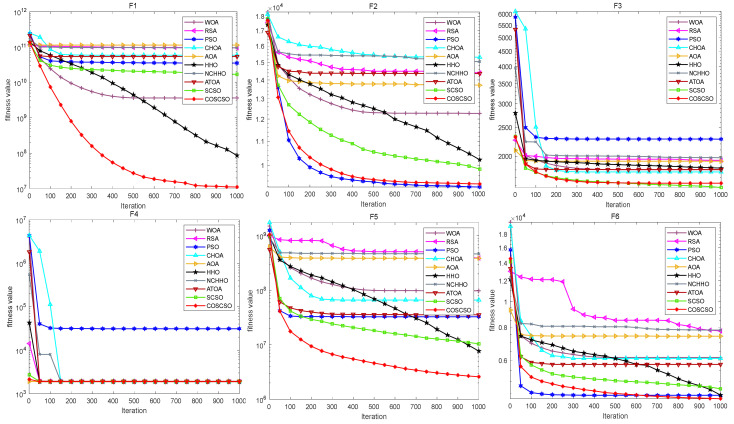
Convergence curves of COSCSO with other algorithms (50-dimensional CEC2020).

**Figure 13 biomimetics-08-00191-f013:**
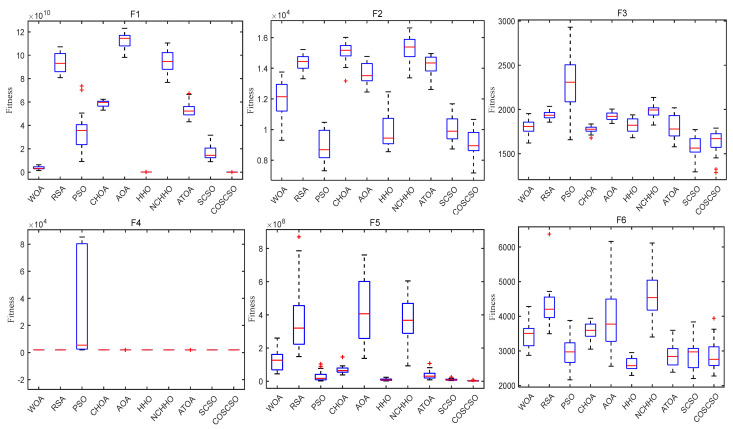
Box plots of COSCSO with other algorithms (50-dimensional CEC2020).

**Figure 14 biomimetics-08-00191-f014:**
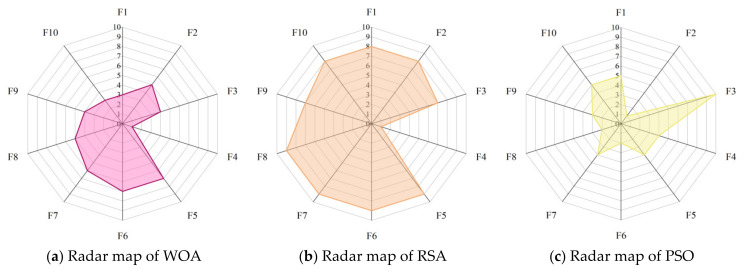
Radar maps of COSCSO with other algorithms (50-dimensional CEC2020).

**Figure 15 biomimetics-08-00191-f015:**
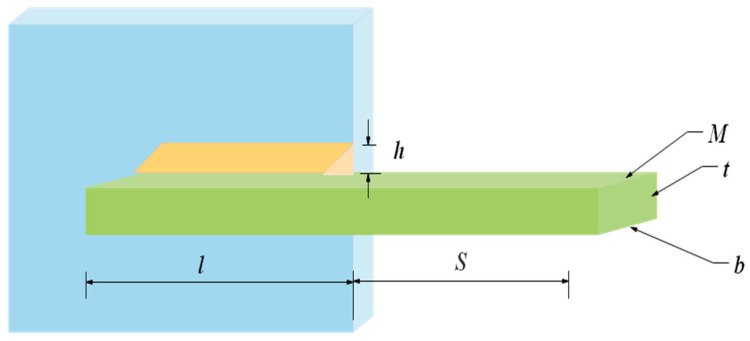
Welded beam design problem.

**Figure 16 biomimetics-08-00191-f016:**
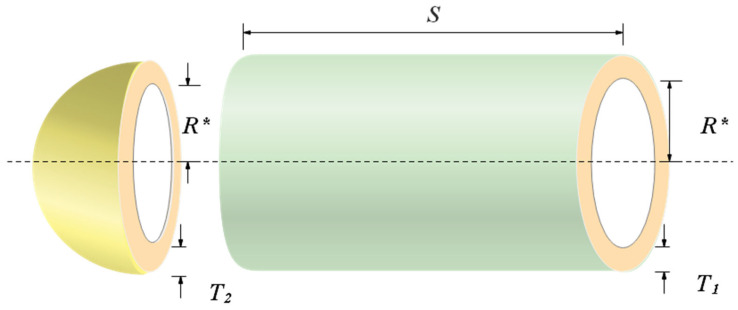
Pressure vessel design problem.

**Figure 17 biomimetics-08-00191-f017:**
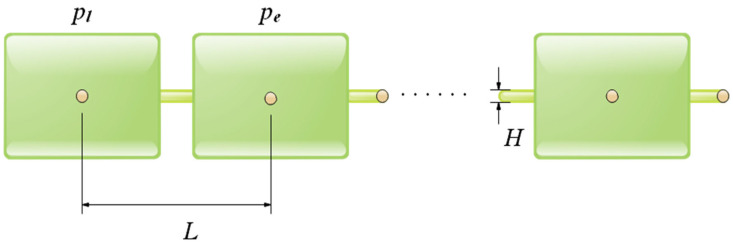
Gas transmission compressor design problem.

**Figure 18 biomimetics-08-00191-f018:**
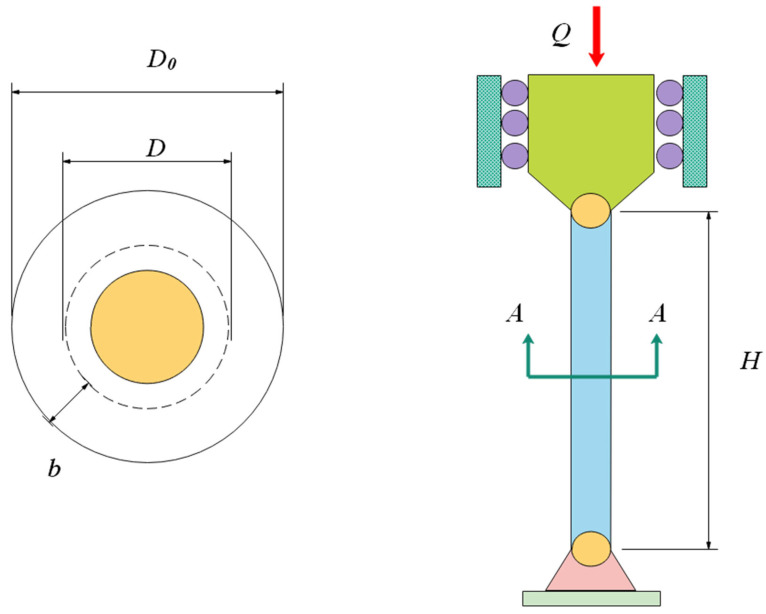
Tubular column design problem.

**Figure 19 biomimetics-08-00191-f019:**
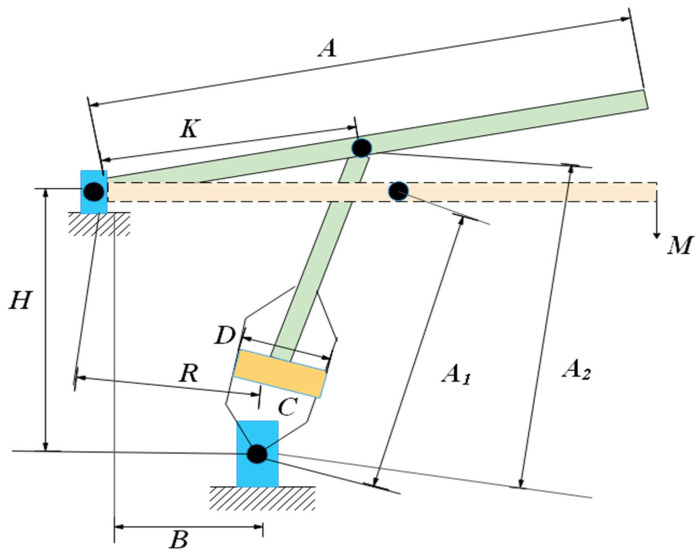
Piston lever design.

**Table 1 biomimetics-08-00191-t001:** Pseudo-code of COSCSO algorithm.

Algorithm: The COSCSO algorithm.
Initialize individuals ***X***_i_ (I = 1,2,∖,N)
Calculate the fitness values for all individuals.
1: **While** (*t* < *T*)
2: Update the parameters like Se, re, Re;
3: **For** each individual
4: Get a random angle based on Roulette Wheel Selection (0° ≤ *θ* ≤ 360°);
5: **If** $(\left|R_{e}\right|\leq 1$)
6: Update the individual position in conformity with Equation (15);
7: **Else**
8: Update the individual position in conformity with Equation (6);
9: **End**
10: Calculate the fitness values of individuals, Produce the *X_best_*(t);
11: Produce the *X*^*^*_best_*(t) in conformity with Equation (16);
12: Calculate the fitness, Update the *X_best_*(t);
13: **End**
14: t=t++
15: **End**

**Table 2 biomimetics-08-00191-t002:** Parameters setting in traditional classical algorithms.

Algorithms	Parameters Name	Parameters Values
PSO	Self-learning factor *o_1_* Group learning factor *o_2_* Inertia weight ω	0.5 0.5 0.8
RSA	Sensitive parameter *α* Control parameter *β*	0.1 0.05
BWO	Balance factor *B_f_*	(0, 1)
DO	Adaptive parameter *α*	[0, 1]
AOA	Control parameter σ Sensitive parameter *v*	0.499 0.5
HHO	Initial energy *E*_0_	[−1, 1]
ATOA	Sensitive parameter *α*	5
NCHHO	Control parameter *c* Control parameter *a*_1_	[0, 2] 4
WOA	Control parameter *m* Constant *n*	Linearly decreases from 2 to 0 1
CHOA	parameter *f*	Linearly decreases from 2 to 0

**Table 3 biomimetics-08-00191-t003:** Comparison results on functions of CEC2017 (Bold type is the optimal value).

F	Results	Algorithms									
		PSO	RSA	BWO	DO	AOA	HHO	NCHHO	ATOA	SCSO	COSCSO
F1	Mean	8.08E+09	4.64E+10	4.76E+10	1.67E+09	4.86E+10	1.55E+07	4.37E+10	2.20E+10	3.88E+09	**3.48E+05**
	Std	6.54E+09	6.30E+09	5.54E+09	9.53E+08	1.10E+10	3.75E+06	9.77E+09	8.28E+09	1.74E+09	**7.03E+05**
	Rank	5	8	9	3	10	2	7	6	4	1
	*p*	6.80E-08	6.80E-08	6.80E-08	6.80E-08	6.80E-08	6.80E-08	6.80E-08	6.80E-08	6.80E-08	
F3	Mean	5.84E+04	7.86E+04	7.69E+04	8.56E+04	7.68E+04	2.13E+04	8.96E+04	8.14E+04	4.37E+04	**1.11E+04**
	Std	2.57E+04	5.33E+03	4.27E+03	5.64E+03	7.18E+03	5.48E+03	4.00E+03	9.29E+03	7.06E+03	**3.26E+03**
	Rank	4	7	5	9	6	2	10	8	3	1
	*p*	9.17E-08	6.80E-08	6.80E-08	6.80E-08	6.80E-08	1.38E-06	6.80E-08	6.80E-08	6.80E-08	
F4	Mean	1.35E+03	8.84E+03	1.15E+04	6.91E+02	1.28E+04	5.41E+02	1.14E+04	2.43E+03	7.47E+02	**5.05E+02**
	Std	9.28E+02	2.22E+03	1.12E+02	1.02E+02	3.34E+03	2.74E+01	3.18E+03	1.40E+03	3.01E+02	**1.96E+01**
	Rank	5	7	9	3	10	2	8	6	4	1
	*p*	6.80E-08	6.80E-08	6.80E-08	1.23E-07	6.80E-08	1.04E-04	6.80E-08	6.80E-08	2.56E-07	
F5	Mean	**6.99E+02**	9.14E+02	9.11E+02	7.84E+02	8.92E+02	7.39E+02	8.93E+02	8.38E+02	7.32E+02	7.24E+02
	Std	4.04E+01	2.62E+01	**2.12E+01**	4.84E+01	2.80E+01	2.45E+01	4.40E+01	3.27E+01	5.03E+01	4.88E+01
	Rank	1	10	9	5	7	4	8	6	3	2
	*p*	**1.08E-01**	6.80E-08	6.80E-08	9.21E-04	6.80E-08	**1.33E-01**	1.23E-07	3.94E-07	**5.79E-01**	
F6	Mean	**6.49E+02**	6.87E+02	6.86E+02	6.68E+02	6.74E+02	6.64E+02	6.83E+02	6.65E+02	6.60E+02	6.59E+02
	Std	8.98E+00	6.15E+00	**3.79E+00**	4.16E+00	5.69E+00	5.90E+00	7.64E+00	9.24E+00	9.60E+00	8.97E+00
	Rank	1	10	9	6	7	4	8	5	3	2
	*p*	1.79E-04	6.80E-08	6.80E-08	1.16E-04	7.95E-07	**1.08E-01**	7.90E-08	2.56E-02	**9.46E-01**	
F7	Mean	1.15E+03	1.38E+03	1.36E+03	1.25E+03	1.33E+03	1.29E+03	1.35E+03	1.24E+03	1.13E+03	**1.12E+03**
	Std	1.58E+02	**3.95E+01**	4.66E+01	1.03E+02	6.59E+01	7.03E+01	7.41E+01	6.09E+01	9.85E+01	1.14E+02
	Rank	3	10	9	5	7	6	8	4	2	1
	*p*	**6.17E-01**	6.80E-08	9.17E-08	3.38E-04	9.13E-07	2.60E-05	5.23E-07	3.05E-04	**7.35E-01**	
F8	Mean	9.81E+02	1.13E+03	1.13E+03	1.01E+03	1.10E+03	9.72E+02	1.13E+03	1.09E+03	9.91E+02	**9.71E+02**
	Std	4.46E+01	**2.16E+01**	1.19E+01	2.50E+01	2.29E+01	2.76E+01	3.60E+01	3.31E+01	3.16E+01	3.56E+01
	Rank	3	9	8	5	7	2	10	6	4	1
	*p*	**5.43E-01**	6.80E-08	6.80E-08	3.05E-04	6.80E-08	**8.82E-01**	6.80E-08	7.90E-08	**6.79E-02**	
F9	Mean	5.62E+03	1.01E+04	1.06E+04	6.90E+03	6.95E+03	6.88E+03	8.88E+03	1.24E+04	5.58E+03	**5.11E+03**
	Std	2.29E+03	9.13E+02	6.65E+02	6.96E+02	1.12E+03	8.05E+02	1.54E+03	3.16E+03	8.87E+02	**3.22E+02**
	Rank	3	8	9	5	6	4	7	10	2	1
	*p*	**6.17E-01**	6.80E-08	6.80E-08	6.80E-08	1.10E-05	7.95E-07	6.80E-08	6.80E-08	6.87E-04	
F10	Mean	5.48E+03	8.25E+03	8.55E+03	5.75E+03	7.44E+03	5.71E+03	8.26E+03	7.40E+03	5.88E+03	**5.33E+03**
	Std	6.40E+02	3.65E+02	**3.58E+02**	4.41E+02	5.05E+02	5.50E+02	6.11E+02	5.96E+02	7.38E+02	8.03E+02
	Rank	2	8	10	4	7	3	9	6	5	1
	*p*	**5.25E-01**	6.80E-08	6.80E-08	2.75E-02	2.22E-07	9.09E-02	9.17E-08	6.01E-07	**2.75E-02**	
F11	Mean	1.55E+03	8.06E+03	6.60E+03	2.52E+03	7.76E+03	1.28E+03	9.79E+03	7.20E+03	1.73E+03	**1.27E+03**
	Std	1.88E+02	2.79E+03	6.51E+02	6.82E+02	2.19E+03	**4.84E+01**	2.43E+03	4.81E+03	3.10E+02	6.12E+01
	Rank	3	9	6	5	8	2	10	7	4	1
	*p*	2.96E-07	6.80E-08	6.80E-08	6.80E-08	6.80E-08	**6.95E-01**	6.80E-08	6.80E-08	9.17E-08	
F12	Mean	6.49E+08	1.38E+10	9.76E+09	8.43E+07	1.16E+10	1.98E+07	1.76E+07	7.35E+09	1.40E+08	**6.89E+06**
	Std	6.66E+08	3.54E+09	2.11E+09	7.05E+07	2.96E+09	1.48E+07	1.32E+07	2.93E+09	1.77E+08	**8.68E+06**
	Rank	6	10	8	4	9	3	2	7	5	1
	*p*	1.66E-07	6.80E-08	6.80E-08	1.92E-07	6.80E-08	5.63E-04	6.80E-08	6.80E-08	1.06E-07	
F13	Mean	4.21E+08	1.08E+10	5.55E+09	1.29E+06	7.67E+09	4.45E+05	4.30E+09	3.00E+08	1.44E+07	**1.38E+05**
	Std	9.20E+08	4.05E+09	1.62E+09	4.16E+06	4.65E+09	1.42E+05	2.08E+09	3.34E+08	2.67E+07	**1.03E+05**
	Rank	6	10	8	3	9	2	7	5	4	1
	*p*	**1.44E-04**	6.80E-08	6.80E-08	3.97E-03	6.80E-08	1.05E-06	6.80E-08	6.80E-08	2.07E-02	
F14	Mean	1.25E+05	7.35E+06	2.27E+06	1.58E+06	1.15E+06	2.85E+05	6.36E+06	1.67E+06	4.34E+05	**4.44E+04**
	Std	1.89E+05	9.53E+06	1.02E+06	9.04E+05	1.33E+06	2.30E+05	5.46E+06	1.57E+06	7.20E+05	**3.31E+04**
	Rank	2	10	8	6	5	3	9	7	4	1
	*p*	1.23E-02	6.80E-08	6.80E-08	6.80E-08	2.36E-06	5.90E-05	6.80E-08	7.58E-06	1.35E-03	
F15	Mean	7.03E+04	6.22E+08	2.15E+08	6.75E+04	2.31E+06	5.44E+04	1.27E+08	1.41E+04	2.29E+05	**4.53E+04**
	Std	4.11E+04	3.12E+08	1.08E+08	7.62E+04	1.02E+07	3.43E+04	2.27E+08	**1.07E+04**	5.94E+05	3.22E+04
	Rank	5	10	9	4	7	3	8	1	6	2
	*p*	1.55E-02	6.80E-08	6.80E-08	**3.23E-01**	3.06E-03	**3.79E-01**	6.80E-08	9.75E-06	1.81E-01	
F16	Mean	3.15E+03	5.64E+03	5.24E+03	3.50E+03	4.78E+03	3.17E+03	5.00E+03	3.54E+03	3.12E+03	**3.08E+03**
	Std	3.54E+02	1.47E+03	5.18E+02	4.37E+02	7.45E+02	3.86E+02	6.24E+02	3.41E+02	3.66E+02	**3.36E+02**
	Rank	3	10	9	5	7	4	8	6	2	1
	*p*	**3.79E-01**	6.80E-08	6.80E-08	3.64E-03	1.06E-07	**4.90E-01**	6.80E-08	5.09E-04	**5.08E-01**	
F17	Mean	2.53E+03	5.34E+03	3.74E+03	2.56E+03	3.31E+03	2.55E+03	3.08E+03	2.69E+03	2.47E+03	**2.44E+03**
	Std	2.74E+02	3.62E+03	3.82E+02	3.21E+02	5.48E+02	3.57E+02	5.03E+02	**1.44E+02**	2.82E+02	2.86E+02
	Rank	3	10	9	5	8	4	7	6	2	1
	*p*	**5.61E-01**	6.80E-08	7.90E-08	**2.98E-01**	9.13E-07	**4.41E-01**	3.29E-05	2.14E-03	**7.15E-01**	
F18	Mean	2.34E+06	3.35E+07	2.70E+07	6.72E+06	1.19E+07	1.86E+06	5.49E+07	4.21E+06	1.20E+06	**9.94E+05**
	Std	6.10E+06	2.73E+07	1.40E+07	5.32E+06	8.24E+06	1.76E+06	4.93E+07	3.08E+06	1.11E+06	**1.01E+06**
	Rank	4	9	8	6	7	3	10	5	2	1
	*p*	**9.03E-01**	6.80E-08	6.80E-08	8.60E-06	1.92E-07	**7.64E-02**	2.96E-07	1.25E-05	**2.85E-01**	
F19	Mean	1.62E+07	6.19E+08	3.07E+08	1.14E+06	1.79E+06	3.46E+05	2.20E+08	3.41E+04	4.23E+06	**2.44E+05**
	Std	4.56E+07	2.77E+08	1.30E+08	9.55E+05	7.77E+04	2.09E+05	2.70E+08	4.54E+04	9.10E+06	**4.08E+05**
	Rank	7	10	9	4	5	3	8	1	6	2
	*p*	**7.35E-01**	6.80E-08	6.80E-08	3.75E-04	9.13E-07	9.05E-03	6.80E-08	7.58E-06	8.29E-05	
F20	Mean	2.72E+03	2.95E+03	2.91E+03	2.99E+03	2.78E+03	2.72E+03	2.99E+03	2.89E+03	2.69E+03	**2.66E+03**
	Std	2.65E+02	1.31E+02	**1.03E+02**	2.56E+02	2.17E+02	1.96E+02	2.19E+02	1.91E+02	1.83E+02	1.62E+02
	Rank	4	8	7	10	5	3	9	6	2	1
	*p*	**4.90E-01**	5.87E-06	2.69E-06	7.41E-05	**6.79E-02**	**2.73E-01**	2.92E-05	6.87E-04	**7.35E-01**	
F21	Mean	**2.49E+03**	2.69E+03	2.71E+03	2.55E+03	2.65E+03	2.55E+03	2.70E+03	2.62E+03	2.51E+03	2.54E+03
	Std	4.39E+01	4.15E+01	3.47E+01	1.05E+02	4.79E+01	**5.16E+01**	5.23E+01	4.29E+01	3.48E+01	4.51E+01
	Rank	1	8	10	5	7	4	9	6	2	3
	*p*	4.16E-04	6.80E-08	6.80E-08	**2.18E-01**	2.06E-06	**9.89E-01**	1.66E-07	7.41E-05	1.93E-02	
F22	Mean	6.20E+03	8.35E+03	8.30E+03	5.30E+03	8.25E+03	5.74E+03	9.38E+03	7.63E+03	3.49E+03	**3.02E+03**
	Std	1.79E+03	1.06E+03	**6.84E+02**	2.65E+03	1.08E+03	2.06E+03	1.04E+03	2.21E+03	1.30E+03	1.77E+03
	Rank	5	8	7	3	9	4	10	6	2	1
	*p*	1.41E-05	3.42E-07	1.66E-07	9.75E-06	3.42E-07	2.30E-05	1.43E-07	1.38E-06	1.29E-04	
F23	Mean	2.97E+03	3.25E+03	3.28E+03	3.07E+03	3.51E+03	3.13E+03	3.67E+03	3.07E+03	**2.93E+03**	2.92E+03
	Std	8.40E+01	7.39E+01	**6.25E+01**	1.48E+02	1.45E+02	1.40E+02	1.39E+02	6.62E+01	4.15E+01	6.87E+01
	Rank	3	7	8	5	9	6	10	4	2	1
	*p*	**8.59E-02**	6.80E-08	6.80E-08	2.47E-04	6.80E-08	3.07E-06	6.80E-08	1.05E-06	**8.39E-01**	
F24	Mean	3.16E+03	3.47E+03	3.52E+03	3.19E+03	3.80E+03	3.11E+03	3.80E+03	3.28E+03	3.09E+03	**3.08E+03**
	Std	7.63E+01	1.59E+02	7.24E+01	7.65E+01	2.48E+02	**1.84E+01**	2.38E+02	6.80E+01	6.91E+01	6.44E+01
	Rank	4	7	8	5	10	3	9	6	2	1
	*p*	4.32E-03	6.80E-08	6.80E-08	2.60E-05	6.80E-08	**1.99E-01**	6.80E-08	1.06E-07	**6.95E-01**	
F25	Mean	3.17E+03	4.85E+03	4.31E+03	3.03E+03	5.28E+03	2.92E+03	4.60E+03	3.43E+03	3.09E+03	**2.93E+03**
	Std	3.42E+02	6.52E+02	1.63E+02	5.05E+01	8.73E+02	2.30E+01	4.43E+02	2.85E+02	7.83E+01	**2.17E+01**
	Rank	5	9	7	3	10	2	8	6	4	1
	*p*	9.17E-08	6.80E-08	6.80E-08	6.80E-08	6.80E-08	**6.55E-01**	6.80E-08	6.80E-08	7.90E-08	
F26	Mean	6.73E+03	9.68E+03	1.03E+04	6.96E+03	1.04E+04	7.56E+03	1.05E+04	8.28E+03	6.63E+03	**6.52E+03**
	Std	8.80E+02	8.20E+02	3.43E+02	1.76E+03	**1.02E+03**	1.30E+03	1.11E+03	1.13E+03	1.41E+03	1.86E+03
	Rank	3	7	8	4	9	5	10	6	2	1
	*p*	9.46E-01	3.94E-07	6.80E-08	**2.73E-01**	1.06E-07	4.68E-02	1.06E-07	1.23E-03	**9.89E-01**	
F27	Mean	3.33E+03	3.89E+03	3.90E+03	3.39E+03	4.39E+03	3.42E+03	4.53E+03	3.42E+03	3.36E+03	**3.32E+03**
	Std	**7.58E+01**	5.05E+02	1.34E+02	7.83E+01	3.28E+02	1.14E+02	5.01E+02	8.17E+01	7.70E+01	8.83E+01
	Rank	2	7	8	5	9	4	10	6	3	1
	*p*	**3.94E-01**	3.42E-07	6.80E-08	2.22E-04	6.80E-08	1.61E-04	7.90E-08	3.71E-05	2.23E-02	
F28	Mean	4.26E+03	5.89E+03	6.20E+03	3.47E+03	6.89E+03	3.29E+03	6.31E+03	4.23E+03	3.53E+03	**3.26E+03**
	Std	8.73E+02	9.02E+02	2.92E+02	5.79E+01	9.24E+02	**1.77E+01**	8.83E+02	4.62E+02	1.37E+02	2.50E+01
	Rank	6	7	8	3	10	2	9	5	4	1
	*p*	6.80E-08	6.80E-08	6.80E-08	6.80E-08	6.80E-08	1.44E-04	6.80E-08	6.80E-08	6.80E-08	
F29	Mean	4.63E+03	7.40E+03	6.51E+03	4.88E+03	7.12E+03	4.60E+03	6.63E+03	4.70E+03	4.57E+03	**4.55E+03**
	Std	4.96E+02	3.56E+03	5.43E+02	3.71E+02	1.06E+03	4.09E+02	9.81E+02	3.68E+02	**3.61E+02**	4.22E+02
	Rank	3	10	7	6	9	4	8	5	2	1
	*p*	**5.43E-01**	1.06E-07	1.06E-07	8.35E-03	6.80E-08	**5.25E-01**	1.23E-07	**2.08E-01**	**5.25E-01**	
F30	Mean	6.67E+07	2.76E+09	8.24E+08	1.11E+07	1.14E+09	3.33E+06	6.27E+08	5.55E+07	8.36E+07	**1.95E+06**
	Std	2.56E+08	9.66E+08	3.24E+08	5.53E+06	1.01E+09	2.02E+06	4.99E+08	5.30E+07	9.62E+06	**1.41E+06**
	Rank	5	10	8	3	9	2	7	4	6	1
	*p*	3.97E-03	6.80E-08	6.80E-08	2.22E-07	6.80E-08	2.39E-02	6.80E-08	3.42E-07	1.58E-06	
Mean rank		3.6897	8.3793	8.4828	4.7931	7.8621	3.2759	8.3793	5.8276	3.3103	1.2414
Result		4	8	9	5	7	2	8	6	3	1
+/=/−		2/13/14	0/0/29	0/0/29	0/4/25	0/1/28	0/13/16	0/0/29	1/1/27	0/12/17	

**Table 4 biomimetics-08-00191-t004:** Comparison results on functions of 30-dimensional CEC2020.

F	Results	Algorithms									
		WOA	RSA	PSO	CHOA	AOA	HHO	NCHHO	ATOA	SCSO	COSCSO
F1	Mean	5.27E+08	5.14E+10	6.81E+09	2.83E+10	4.80E+10	1.72E+07	4.31E+10	2.26E+10	4.65E+09	**6.08E+05**
	Std	3.25E+08	8.10E+09	5.60E+09	5.05E+09	7.81E+09	4.11E+06	8.72E+09	7.55E+09	3.17E+09	**8.65E+05**
	Rank	3	10	5	7	9	2	8	6	4	1
	*p*	6.80E-08	6.80E-08	6.80E-08	6.80E-08	6.80E-08	6.80E-08	6.80E-08	6.80E-08	6.80E-08	
F2	Mean	6.46E+03	7.65E+03	5.48E+03	8.01E+03	7.20E+03	5.44E+03	8.44E+03	7.32E+03	6.15E+03	**5.22E+03**
	Std	8.01E+02	**4.01E+02**	6.19E+02	6.76E+02	5.69E+02	6.31E+02	6.41E+02	5.48E+02	7.64E+02	6.89E+02
	Rank	5	8	3	9	6	2	10	7	4	1
	*p*	9.75E-06	6.80E-08	**3.10E-01**	1.06E-07	2.22E-07	3.65E-01	2.36E-06	1.43E-07	5.63E-04	
F3	Mean	1.27E+03	1.38E+03	1.14E+03	1.26E+03	1.36E+03	1.27E+03	1.33E+03	1.26E+03	1.15E+03	**1.14E+03**
	Std	9.35E+01	**2.98E+01**	1.58E+02	1.83E+01	6.55E+01	8.55E+01	7.13E+01	6.16E+01	6.24E+01	1.01E+02
	Rank	7	10	2	5	9	6	8	4	3	1
	*p*	4.60E-04	7.90E-08	**2.85E-01**	1.04E-04	7.90E-08	1.63E-03	5.23E-07	2.39E-02	**7.35E-01**	
F4	Mean	**1.90E+03**	**1.90E+03**	1.95E+03	**1.90E+03**	**1.90E+03**	**1.90E+03**	**1.90E+03**	1.91E+03	**1.90E+03**	**1.90E+03**
	Std	0.00E+00	**0.00E+00**	3.04E+01	**0.00E+00**	**0.00E+00**	**0.00E+00**	**0.00E+00**	5.47E+00	**0.00E+00**	**0.00E+00**
	Rank	1	1	3	1	1	1	1	2	1	1
	*p*	NaN	NaN	8.01E-09	NaN	NaN	NaN	NaN	8.01E-09	NaN	
F5	Mean	9.80E+06	6.41E+07	4.72E+06	2.53E+07	7.13E+07	2.47E+06	8.03E+07	9.09E+06	2.96E+06	**1.26E+06**
	Std	6.58E+06	4.36E+07	6.21E+06	1.76E+07	4.66E+07	1.62E+06	4.61E+07	6.31E+06	3.64E+06	**1.00E+06**
	Rank	6	8	4	7	9	2	10	5	3	1
	*p*	6.01E-07	6.80E-08	2.80E-03	6.80E-08	6.80E-08	1.14E-02	6.80E-08	6.80E-08	**1.90E-01**	
F6	Mean	3.43E+03	4.49E+03	2.94E+03	3.43E+03	3.99E+03	**2.59E+03**	4.57E+03	2.78E+03	2.70E+03	2.67E+03
	Std	5.77E+02	6.81E+02	3.37E+02	3.51E+02	1.25E+03	3.41E+02	8.14E+02	**2.95E+02**	3.61E+02	3.09E+02
	Rank	7	9	5	6	8	1	10	4	3	2
	*p*	5.17E-06	6.80E-08	1.93E-02	2.06E-06	3.99E-06	**2.85E-01**	6.80E-08	**2.98E-01**	**9.89E-01**	
F7	Mean	5.60E+06	4.32E+07	8.30E+05	7.52E+06	2.25E+07	6.40E+05	3.22E+07	2.99E+06	2.44E+06	**3.42E+05**
	Std	3.92E+06	2.21E+07	6.16E+05	4.42E+06	1.57E+07	4.88E+05	2.16E+07	2.12E+06	3.40E+06	**3.36E+05**
	Rank	6	10	3	7	8	2	9	5	4	1
	*p*	2.22E-07	6.80E-08	1.95E-03	1.66E-07	6.80E-08	9.79E-03	6.80E-08	2.96E-07	1.48E-03	
F8	Mean	7.11E+03	8.74E+03	6.36E+03	8.19E+03	8.69E+03	5.61E+03	8.94E+03	8.83E+03	4.12E+03	**3.26E+03**
	Std	1.56E+03	8.15E+02	1.74E+03	1.32E+03	**1.12E+03**	2.29E+03	1.37E+03	1.00E+03	2.02E+03	2.00E+03
	Rank	5	8	4	6	7	3	10	9	2	1
	*p*	7.58E-06	1.92E-07	1.60E-05	1.20E-06	3.94E-07	5.26E-05	3.42E-07	2.22E-07	3.75E-04	
F9	Mean	3.24E+03	3.48E+03	3.12E+03	3.31E+03	3.78E+03	3.41E+03	3.91E+03	3.31E+03	3.09E+03	**3.07E+03**
	Std	8.48E+01	2.20E+02	**1.56E+01**	3.24E+01	1.75E+02	1.29E+02	2.66E+02	1.04E+02	5.90E+01	9.11E+01
	Rank	4	8	3	6	9	7	10	5	2	**1**
	*p*	5.17E-06	6.80E-08	2.34E-03	1.66E-07	6.80E-08	1.43E-07	6.80E-08	7.95E-07	**2.08E-01**	
F10	Mean	3.06E+03	4.72E+03	3.24E+03	4.54E+03	5.06E+03	2.94E+03	4.50E+03	3.56E+03	3.09E+03	**2.93E+03**
	Std	5.77E+01	5.67E+02	2.41E+02	4.58E+02	7.40E+02	**2.29E+01**	6.05E+02	3.13E+02	9.65E+01	2.41E+01
	Rank	3	9	5	8	10	2	7	6	4	1
	*p*	6.80E-08	6.80E-08	6.80E-08	6.80E-08	6.80E-08	**2.50E-01**	6.80E-08	6.80E-08	7.90E-08	
Mean rank		4.7	8.1	3.7	6.2	7.6	2.8	8.3	5.3	3.0	1.1
Result		5	9	4	7	8	2	10	6	3	1
+/=/−		0/1/9	0/1/9	0/2/8	0/1/9	0/1/9	0/3/7	0/1/9	0/1/9	0/4/6	

**Table 5 biomimetics-08-00191-t005:** Comparison results on functions of 50-dimensional CEC2020.

F	Results	Algorithms									
		WOA	RSA	PSO	CHOA	AOA	HHO	NCHHO	ATOA	SCSO	COSCSO
F1	Mean	3.63E+09	9.43E+10	3.50E+10	5.87E+10	1.13E+11	8.59E+07	9.44E+10	5.35E+10	1.69E+10	**1.11E+07**
	Std	1.31E+09	8.75E+09	1.62E+10	2.76E+09	6.18E+09	1.88E+07	9.67E+09	7.04E+09	6.50E+09	**1.06E+07**
	Rank	3	8	5	7	10	2	9	6	4	1
	*p*	6.80E-08	6.80E-08	6.80E-08	6.80E-08	6.80E-08	6.80E-08	6.80E-08	6.80E-08	6.80E-08	
F2	Mean	1.20E+04	1.44E+04	**8.93E+03**	1.51E+04	1.37E+04	9.90E+03	1.52E+04	1.42E+04	1.01E+04	9.15E+03
	Std	1.15E+03	**5.84E+02**	1.01E+03	6.70E+02	6.81E+02	1.09E+03	8.54E+02	7.02E+02	8.38E+02	8.92E+02
	Rank	5	8	1	9	6	3	10	7	4	2
	*p*	3.42E-07	6.80E-08	**3.94E-01**	6.80E-08	6.80E-08	3.60E-02	6.80E-08	6.80E-08	3.97E-03	
F3	Mean	1.80E+03	1.94E+03	2.28E+03	1.77E+03	1.92E+03	1.82E+03	1.98E+03	1.80E+03	1.57E+03	**1.62E+03**
	Std	8.51E+01	**4.61E+01**	3.32E+02	5.16E+01	4.70E+01	7.88E+01	7.71E+01	1.23E+02	1.17E+02	1.43E+02
	Rank	4	7	10	3	8	6	9	5	1	2
	*p*	3.29E-05	6.80E-08	6.92E-07	1.60E-05	6.80E-08	7.58E-06	6.80E-08	4.60E-04	**7.64E-02**	
F4	Mean	**1.90E+03**	**1.90E+03**	3.13E+04	**1.90E+03**	**1.90E+03**	**1.90E+03**	**1.90E+03**	1.93E+03	**1.90E+03**	**1.90E+03**
	Std	**0.00E+00**	**0.00E+00**	3.83E+04	**0.00E+00**	8.80E-06	**0.00E+00**	**0.00E+00**	1.09E+01	**0.00E+00**	**0.00E+00**
	Rank	1	1	4	1	2	1	1	3	1	1
	*p*	NaN	NaN	8.01E-09	NaN	6.68E-05	NaN	NaN	8.01E-09	NaN	
F5	Mean	1.26E+08	3.85E+08	3.00E+07	6.87E+07	4.35E+08	9.20E+06	3.73E+08	3.81E+07	9.72E+06	**2.49E+06**
	Std	6.57E+07	2.20E+08	3.03E+07	2.41E+07	1.87E+08	5.57E+06	1.41E+08	2.48E+07	5.11E+06	**1.55E+06**
	Rank	7	9	4	6	10	2	8	5	3	1
	*p*	6.80E-08	6.80E-08	7.95E-07	6.80E-08	6.80E-08	2.36E-06	6.80E-08	6.80E-08	3.42E-07	
F6	Mean	6.15E+03	7.76E+03	4.43E+03	6.11E+03	7.41E+03	4.44E+03	7.81E+03	5.80E+03	4.69E+03	**4.30E+03**
	Std	7.63E+02	5.94E+02	4.02E+02	**4.00E+02**	1.44E+03	5.78E+02	1.36E+03	8.39E+02	7.27E+02	7.90E+02
	Rank	7	9	2	6	8	3	10	5	4	1
	*p*	1.58E-06	6.80E-08	**3.94E-01**	1.23E-07	7.90E-08	**4.57E-01**	6.80E-08	1.10E-05	**9.62E-02**	
F7	Mean	1.59E+07	8.32E+07	8.53E+06	2.35E+07	5.78E+07	5.29E+06	9.07E+07	1.73E+07	4.30E+06	**1.40E+06**
	Std	8.79E+06	5.05E+07	6.39E+06	6.08E+06	3.21E+07	2.80E+06	4.58E+07	9.63E+06	3.82E+06	**9.03E+05**
	Rank	6	9	4	7	8	3	10	5	2	1
	*p*	4.54E-06	6.80E-08	2.06E-06	6.80E-08	6.80E-08	2.36E-06	6.80E-08	1.23E-07	1.12E-03	
F8	Mean	1.36E+04	1.71E+04	1.04E+04	1.71E+04	1.56E+04	1.12E+04	1.64E+04	1.59E+04	1.10E+04	**1.00E+04**
	Std	1.02E+03	3.92E+02	8.71E+02	5.94E+02	6.30E+02	7.42E+02	6.67E+02	6.78E+02	**2.02E+03**	2.06E+03
	Rank	5	9	2	10	6	4	8	7	3	1
	*p*	1.43E-07	6.80E-08	**9.25E-01**	6.80E-08	6.80E-08	1.93E-02	6.80E-08	6.80E-08	3.37E-02	
F9	Mean	3.80E+03	4.17E+03	3.65E+03	4.05E+03	4.88E+03	4.25E+03	4.86E+03	3.88E+03	3.44E+03	**3.44E+03**
	Std	1.36E+02	4.42E+02	1.58E+02	4.76E+01	2.48E+02	2.46E+02	4.38E+02	1.66E+02	1.28E+02	**1.27E+02**
	Rank	4	7	3	6	10	8	9	5	2	**1**
	*p*	5.23E-07	6.80E-08	4.68E-05	6.80E-08	6.80E-08	6.80E-08	6.80E-08	1.06E-07	**9.25E-01**	
F10	Mean	3.72E+03	1.27E+04	6.14E+03	1.04E+04	1.58E+04	3.19E+03	1.39E+04	8.21E+03	4.32E+03	**3.12E+03**
	Std	2.48E+02	1.71E+03	1.93E+03	8.38E+02	1.66E+03	3.95E+01	1.78E+03	1.62E+03	4.74E+02	**2.59E+01**
	Rank	3	8	5	7	10	2	9	6	4	1
	*p*	6.80E-08	6.80E-08	6.80E-08	6.80E-08	6.80E-08	5.23E-07	6.80E-08	6.80E-08	6.80E-08	
Mean rank		4.5	7.5	4.0	6.4	7.8	3.4	8.3	5.4	2.8	1.4
Result		5	8	4	7	9	3	10	6	2	1
+/=/−		0/1/9	0/1/9	0/3/6	0/1/9	0/0/10	0/2/8	0/1/9	0/0/10	0/4/6	

**Table 6 biomimetics-08-00191-t006:** The optimal result of welded beam design.

Algorithms	Optimum Variables				Optimum Cost
	*k* _1_	*k* _2_	*k* _3_	*k* _4_	
WOA	0.176794286	3.891577393	9.255735323	0.204661441	1.764910575
AO	0.202106856	3.319686034	9.081046814	0.212260217	1.755925500
SCA	0.165196847	4.294568905	8.967124254	0.210654365	1.792045440
RSA	0.175670680	3.624398648	9.999449030	0.205952534	1.869756941
HS	0.177543770	4.098912342	8.878245918	0.217531907	1.824393253
BWO	0.218273731	3.015636000	9.273401650	0.220363497	1.831589741
HHO	0.201474520	3.378944473	8.960620234	0.218644819	1.719100771
AOA	0.209983002	3.017257244	10.00000000	0.212237971	1.884562862
SCSO	3.267355241	3.267355241	9.035052358	0.205802427	1.696622053
COSCSO	0.205747115	3.252954257	9.036236158	0.205747352	**1.695317058**

**Table 7 biomimetics-08-00191-t007:** Statistical results of welded beam design.

Algorithms	Best	Worst	Mean	Std
WOA	1.764910575	3.268065060	2.360271979	0.437518374
AO	1.755925500	2.621445835	2.074867078	0.197032112
SCA	1.792045440	1.992645879	1.862883723	0.051244352
RSA	1.869756941	27.442208090	3.681496058	5.601754100
HS	1.824393253	3.706355014	2.719342102	0.498207128
BWO	1.831589741	2.952547258	2.251305241	0.284732148
HHO	1.719100771	2.313552829	1.850862237	0.147918368
AOA	1.884562862	2.914716233	2.309549772	0.313092730
SCSO	1.696622053	4.242983833	2.004451488	0.765112742
COSCSO	**1.695317058**	**1.781726391**	**1.713814493**	**0.021561647**

**Table 8 biomimetics-08-00191-t008:** The optimal result of pressure vessel design.

Algorithms	Optimum Variables				Optimum Cost
	*k* _1_	*k* _2_	*k* _3_	*k* _4_	
WOA	0.795338105	0.654521576	40.86741439	192.5123053	6736.71434
AO	0.801045506	0.418921940	41.48734742	185.7712132	6030.23905
HS	0.978539020	0.500973483	49.50650039	103.1850619	6547.72658
RSA	0.972997715	0.512637098	47.36047546	200	6305.50049
SCA	0.854015786	0.520118487	44.11567303	158.4223506	5885.88792
BWO	0.938950393	0.563152725	48.41429245	115.1168137	5885.33405
BSA	0.778164873	0.493776584	40.31964778	199.9995955	7303.71890
AOA	0.793199905	0.458365929	40.34176154	200	5885.31787
SCSO	0.796043593	0.406047822	41.24545087	187.5713345	5885.32021
COSCSO	0.778539806	0.385198905	40.33914125	199.7284276	**5885.31757**

**Table 9 biomimetics-08-00191-t009:** Statistical results of pressure vessel design.

Algorithms	Best	Worst	Mean	Std
WOA	6736.71434	15021.66294	9511.21867	2551.89041
AO	6030.23905	7756.65903	6882.12163	533.52205
HS	6547.72658	12757.34328	8720.20447	1699.11330
RSA	9269.85205	68265.01725	32898.82493	15501.85116
SCA	6518.94915	9160.64385	7511.17067	772.99225
BWO	6772.30663	9518.89193	7538.79379	683.87846
BSA	6200.76520	30037.17906	11444.68588	6484.51516
AOA	6211.62984	18842.91882	11254.75474	3546.15966
SCSO	5956.21327	23310.15051	7614.91057	3715.97323
COSCSO	**5887.02011**	**7318.91872**	**6569.02774**	**517.46245**

**Table 10 biomimetics-08-00191-t010:** The optimal result of gas transmission compressor design.

Algorithms	Optimum Variables			Optimum Cost
	*k* _1_	*k* _2_	*k* _3_	
RSA	54.99999359	1.194623188	25.00083760	2964527.52459
BWO	55	1.193088691	24.53523919	2964639.69065
SOA	53.65973171	1.190449899	24.74449640	2964378.80113
WOA	53.44872314	1.190109928	24.71816871	2964375.49576
SCA	55	1.195189878	24.73268345	2964474.41040
HS	53.38712267	1.189241767	24.74999745	2964380.11714
AO	53.47061054	1.190026373	24.64034062	2964384.51386
AOA	55	1.200762326	24.62935858	2964730.55260
SCSO	53.45101427	1.190109067	24.71872247	2964375.49653
COSCSO	53.44671239	1.190100716	24.71857897	**2964375.49533**

**Table 11 biomimetics-08-00191-t011:** Statistical results of gas transmission compressor design.

Algorithms	Best	Worst	Mean	Std
RSA	2964527.52459	3016290.08652	2978147.32748	14122.03566
BWO	2964639.69065	2985654.73236	2968570.17953	5235.597886
SOA	2964378.80113	2964502.32306	2964430.85626	39.36551384
WOA	2964375.49576	2964376.19841	2964375.57737	0.15661415
SCA	2964474.41040	2965960.51365	2964893.53068	407.6793260
HS	2964380.11714	2972724.66633	2965744.95719	2116.81140
AO	2964384.51386	2974563.67888	2966781.11226	3285.30995
AOA	2964730.55260	2985136.92184	2970250.27062	6037.25127
SCSO	2964375.49653	2987124.04245	2966285.86187	5997.65711
COSCSO	**2964375.49533**	**2964375.49533**	**2964375.49533**	**5.32331E-09**

**Table 12 biomimetics-08-00191-t012:** The optimal result of heat exchanger design.

Algorithms	Optimum Variables								Optimum Cost
	*k* _1_	*k* _2_	*k* _3_	*k* _4_	*k* * _5_ *	*k* * _6_ *	*k_7_*	*k_8_*	
WOA	8.857E+02	4.785E+03	4.419E+03	7.565E+01	3.233E+02	8.793E+01	1.503E+02	4.233E+02	10089.27612
AO	2.657E+03	4.867E+03	6.451E+03	1.227E+02	3.026E+02	1.735E+02	1.903E+02	3.791E+02	13975.23738
HS	2.319E+03	2.737E+03	5.757E+03	2.414E+02	3.254E+02	1.583E+02	3.023E+02	4.252E+02	10812.90588
RSA	1.120E+03	4.272E+03	4.772E+03	7.793E+01	3.003E+02	3.712E+02	1.469E+02	4.051E+02	27223.59476
BSA	2.869E+03	3.093E+03	5.411E+03	1.051E+02	2.840E+02	2.013E+02	1.774E+02	3.838E+02	11372.90767
BWO	4.001E+03	6.410E+03	4.806E+03	1.448E+02	3.183E+02	1.470E+02	1.852E+02	4.132E+02	15217.14817
HHO	1.174E+03	1.000E+03	6.899E+03	1.253E+02	2.241E+02	1.606E+02	2.711E+02	3.241E+02	9073.094947
AOA	5.052E+03	9.071E+03	9.071E+03	3.209E+01	2.132E+02	1.668E+02	2.137E+02	3.069E+02	23194.46529
SCSO	5.119E+02	2.451E+03	4.512E+03	1.640E+02	3.196E+02	2.226E+02	2.442E+02	4.195E+02	7475.073844
COSCSO	1.084E+03	1.103E+03	5.271E+03	1.997E+02	2.891E+02	1.948E+02	3.071E+02	3.891E+02	**7458.396002**

**Table 13 biomimetics-08-00191-t013:** Statistical results of heat exchanger design.

Algorithms	Best	Worst	Mean	Std
WOA	10089.27612	196031.898	48225.06524	52285.7206
AO	13975.23738	178736.7572	42861.35149	37560.63068
HS	10812.90588	124467.6644	60095.18323	29532.60344
RSA	27223.59476	315644.1889	147440.3931	61985.48452
BSA	11372.90767	161404.451	48673.30656	37923.7065
BWO	15217.14817	120323.2952	60086.46685	27717.89898
HHO	9073.094947	77151.91678	21087.5583	15916.02815
AOA	23194.46529	155972.8617	49743.46087	29775.26723
SCSO	7475.073844	212854.4675	30578.91536	12694.59629
COSCSO	**7458.396002**	**33159.19512**	**12647.27427**	**7803.060165**

**Table 14 biomimetics-08-00191-t014:** The optimal result of tubular column design.

Algorithms	Optimum Variables		Optimum Cost
	*k* _1_	*k* _2_	
HS	5.445466505	0.293088136	26.531748937
SOA	5.450525166	0.291989004	26.497685800
RSA	5.518677222	0.288200889	26.624133745
WOA	5.450602426	0.291877924	26.492127934
SCA	5.452775183	0.291738699	26.495248656
CHOA	5.449801688	0.292230441	26.507063312
BWO	5.434579699	0.29571633	26.618680219
AOA	5.427423919	0.303376181	26.991049027
SCSO	5.452249069	0.291622918	26.486505805
COSCSO	5.452180739	0.291626429	**26.486361480**

**Table 15 biomimetics-08-00191-t015:** Statistical results of tubular column design.

Algorithms	Best	Worst	Mean	Std
HS	26.531748937	29.380491179	27.101286566	0.682159826
SOA	26.497685800	26.651702678	26.546770890	0.043476606
RSA	26.624133745	31.482014142	28.600884307	1.378250449
WOA	26.492127934	28.062545561	27.067123704	0.475587468
SCA	26.495248656	26.911864726	26.663842655	0.103145967
CHOA	26.507063312	26.664362638	26.592798770	0.049578003
BWO	26.618680219	28.701867216	27.247581963	0.524397843
AOA	26.991049027	28.660769408	27.781832446	0.601084705
SCSO	26.486505805	26.488283717	26.487303513	0.000514557
COSCSO	**26.486361480**	**26.486429135**	**26.486367501**	**0.000016150**

**Table 16 biomimetics-08-00191-t016:** The optimal result of piston lever design.

Algorithms	Optimum Variables				Optimum Cost
	*k* _1_	*k* _2_	*k* _3_	*k* _4_	
WOA	0.086602041	2.079956862	4.093800868	119.228613317	8.955469786
SOA	0.050780514	2.044519184	4.083248703	120	8.432800807
MVO	0.05	2.050052620	4.087058119	119.979915073	8.463145662
CHOA	0.073876185	2.081364562	4.095381685	120	8.847198681
SCA	0.05	2.053817149	4.093726603	120	8.505719113
BWO	0.05	2.105295277	4.096763691	120	8.723224918
HHO	0.050019685	2.041900656	4.083032582	119.999227818	8.414435140
AOA	0.271028410	0.271028410	4.162257291	120	57.99492127
SCSO	0.05	2.041589027	4.083079945	120	8.413213831
COSCSO	0.05	2.041513591	4.083027180	120	**8.412698328**

**Table 17 biomimetics-08-00191-t017:** Statistical results of piston lever design.

Algorithms	Best	Worst	Mean	Std
WOA	8.955469786	342.7630967	55.49855602	95.85126869
SOA	8.432800807	9.954077528	43.99002860	0.503774548
MVO	8.463145662	314.1339038	9.861960059	88.25491267
CHOA	8.847198681	11.80905189	0.871394022	0.815693522
SCA	8.505719113	10.23160760	9.410095331	0.542224958
BWO	8.723224918	10.37487240	9.479639241	0.535563793
HHO	8.414435140	411.9250502	96.24528231	122.8529460
AOA	57.99492127	577.6401065	343.1464542	160.6837497
SCSO	8.413213831	56.92881071	10.87687234	10.83963680
COSCSO	**8.412698328**	**8.525870642**	**8.426024955**	**0.028570619**

## Data Availability

The figures utilized to support the findings of this study are embraced in the article.

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
