# Peer review of "An Adaptive Sand Cat Swarm Algorithm Based on Cauchy Mutation and Optimal Neighborhood Disturbance Strategy"

_biomimetics, 2023, doi:10.3390/biomimetics8020191_

Round 1

Reviewer 1 Report

This article proposes an improved version of sand cat swarm optimization algorithm based on the ideas of Cauchy mutation and optimal neighborhood disturbance strategy. The proposed algorithm was designed to solve global optimization problems. The experiments were conducted on several benchmark functions of unimodal, and multimodal, fixed-dimension multimodal, and composite function groups. The results showed the proposed algorithms have better performances in some function.

1.      The first feeling that this paper gave me was that the introduction was too short. I suggest to expand this section with related works which it is requested in paper, so that readers can understand the current research status of the problem, which this work focused on. It is OK to write the contents about related works in introduction, but I don't think introduction need such a big space.

2.      On page 4, equation 2.4 can be rewritten based on the SCSO algorithm Matlab code that they shared. There is a typo in the main paper.

3.      Despite the fact that the simulation results and analysis have been written in a professional manner, the exploration and exploitation of the proposed method (COSCSO) can be extended, I think it would be a good idea to be more bold with this capability.

4.      Several minor problems have been identified. The authors should go through the whole paper to correct them.

5.      English expressions should be improved to make the readers easier.

6.      Summaries the advantages and limitations of the proposed method in practical applications.

English expressions should be improved to make the readers easier.

Reviewer 2 Report

Authors have introduced here a new optimization algorithm using sand cat algorithm and Cauchy mutation. It is an interesting research idea, however the manuscript needs revision before its publication.

1.    In the abstract, define the abbreviations SCSO and COSCSO.

2.    ‘re’ in equation 2.5, and ‘Re’ in equation 2.7 both are in the range of 0 to 2. It did not explain here, how they are different?

3.    ‘Se’ can be zero so ‘re’ can be, therefore the new solution will assign to zero in this case (Eq. 2.6). if zero in not in the search space, the solution you will get will be infeasible. You have to explain it in detail by taking an example, how will you tackle this situation.

4.    In figure 2, you must draw the values of ‘Se’ from Eq. (2.4) and Eq. (3.1) to show the difference.

5.    Pseudo-code in table 1 must be left aligned and indented for clear understanding the steps.

6.     In table 3, F28, what is the reason to highlight two mean?

7.    Some references are in the list without citing in the manuscript.

8.     There are grammatical and typo mistakes.

 Minor editing of English language required
